# Human brain ancestral barcodes

**Darryl Shibata***

Department of Pathology, University of Southern California, Keck School of Medicine, Los Angeles, United States

## eLife Assessment

This study presents a **valuable** conceptual approach that cell lineage can be determined using methylation data. However, the evidence supporting the claims of the author remains **incomplete** after revision. If clarified further as described in the reviews, this approach could be of broad interest to neuroscientists and developmental biologists.

**Abstract** Dynamic CpG methylation 'barcodes' were read from 15,000–21,000 single cells from three human male brains. To overcome sparse sequencing coverage, the barcode had ~31,000 rapidly fluctuating X-chromosome CpG sites (fCpGs), with at least 500 covered sites per cell and at least 30 common sites between cell pairs (average of ~48). Barcodes appear to start methylated and record mitotic ages because excitatory neurons and glial cells that emerge later in development were less methylated. Barcodes are different between most cells, with average pairwise differences (PWDs) of ~0.5 between cells. About 10 cell pairs per million were more closely related with PWDs <0.05. Barcodes appear to record ancestry and reconstruct trees where more related cells had similar phenotypes, albeit some pairs had phenotypic differences. Inhibitory neurons showed more evidence of tangential migration than excitatory neurons, with related cells in different cortical regions. fCpG barcodes become polymorphic during development and can distinguish between thousands of human cells.

***For correspondence:**
dshibata@usc.edu

**Competing interest:** The author declares that no competing interests exist.

## Introduction

Cell lineages outline tissue development. Complete fate maps are possible by direct observation for small organisms such as *C. elegans*, but various elegant experimental fate markers are employed for larger tissues and longer time intervals (*McKenna and Gagnon, 2019*). For human tissues, prior experimental manipulations are impractical, and genomic alterations are employed. Somatic mutations mark subclones and their fates can be reconstructed with DNA sequencing. Recent advances in single-cell technologies potentially allow fate map reconstruction at single-cell resolution.

Here, we show how fCpG DNA methylation (*Gabbutt et al., 2022*) could be used as dynamic barcodes to study human brain development using single-cell epigenomes annotated with their locations and phenotypes (*Tian et al., 2023*). DNA methylation patterns are usually copied between cell divisions, but replication errors are much higher compared to base replication, allowing for more differences between daughter cells. DNA methylation modulates expression and their patterns can be used to infer cell phenotypes (*Tian et al., 2023*; *Loyfer et al., 2023*), but most fCpG sites are present outside of genes or in unexpressed genes. Criteria for our fCpG barcode are as follows: (1) a defined initial pattern in a progenitor cell; (2) polymorphic changes upon cell division; (3) adequate polymorphism to distinguish between most cells; and (4) capability to record ancestry.

The brain has several features that facilitate barcode development and validation. Foremost, there is extensive single-cell methylation data, with thousands of cells annotated by locations and phenotypes (*Tian et al., 2023*). Although billions of cells are present in an adult brain, lineage trees are

**Table 1.** Brain data.

| brain | age | cells | ave fCpG per cell | ave Meth per cell | ave PWD between cell pairs | cell pairs per million | fCpG per pair | closely related pairs (PWD <0.05) | closely related pairs per million | fCpG per nearest neighbor pair* |
|---|---|---|---|---|---|---|---|---|---|---|
| H01 | 42y | 21,836 | 1170 | 0.58 | 0.47 | 197 | 51 | 1385 | 7.0 | 35 |
| H02 | 29y | 16,161 | 1128 | 0.58 | 0.47 | 99 | 48 | 743 | 7.5 | 34 |
| H04 | 58y | 15,434 | 1060 | 0.58 | 0.47 | 73 | 45 | 1078 | 14.8 | 35 |
| | | | | | between brains | | | | | |
| H02-H01 | | | | | 0.47 | 281 | 49 | 785 | 2.8 | 35 |
| H02-H04 | | | | | 0.47 | 178 | 46 | 653 | 3.7 | 34 |
| H04-H01 | | | | | 0.47 | 254 | 47 | 943 | 3.7 | 35 |

compact because growth is largely prenatal. The brain also allows for serial 'stopwatch' barcode sampling because development roughly follows a caudal to rostral pattern, and groups of neurons characteristically stop dividing and differentiate at different times and locations (*Stiles and Jernigan, 2010*). Brainstem neurons emerge early (*Fan et al., 2020*) and their barcodes should most resemble the initial progenitor state, whereas the stopwatch runs longer for excitatory neurons that appear later in development. To facilitate presentation, barcode performance is summarized as follows: The brain fCpG barcode initializes as predominately methylated in the progenitor cell and becomes polymorphic with more diverse barcodes in excitatory neurons that emerge later in development. The barcode becomes sufficiently polymorphic to uniquely distinguish between most sampled brain cells, and barcoded cells organize into lineage trees.

## Results
### fCpG barcode identification

Barcode development was limited by the sparse single-cell data (*Tian et al., 2023*), with <5% of CpG sites sequenced, often with only a single read. Sparse coverage was mitigated with X-chromosome fCpG sites because only a single read can infer a binary (0, 1) state in male individuals. Autosomal CpG sites require at least 2 reads to infer three possible states (0, 0.5, 1). The X-chromosome also simplifies the identification of polymorphic fCpGs because many neurons have different binary states

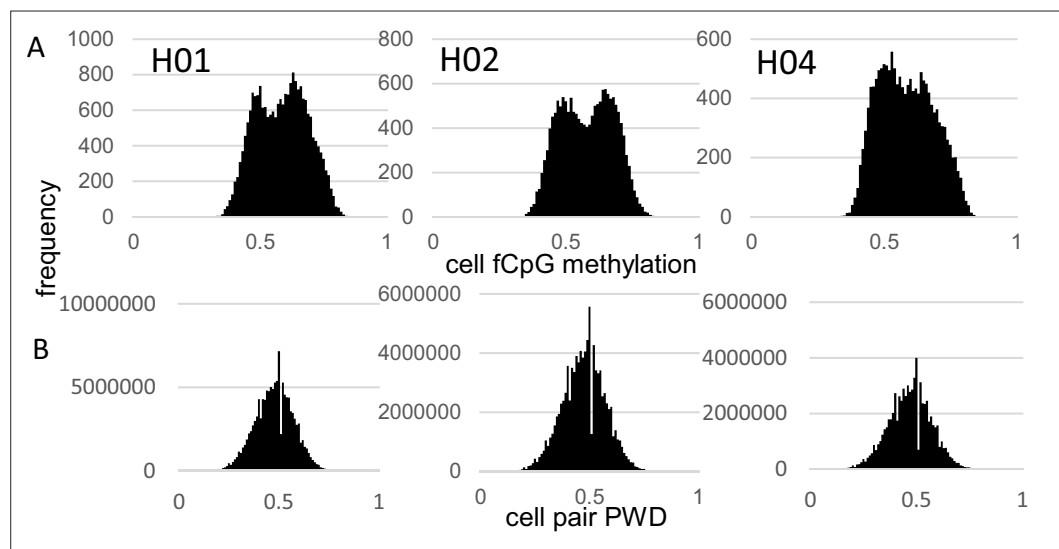

**Figure 1.** Fluctuating CpG (fCpG) barcode methylation 0.5. (**A**) Barcode methylation was variable between cells with averages ~50% (**B**) Most cells had different barcodes with average pairwise differences (PWDs) ~0.5.

if average methylation is between 0.25 and 0.75 in bulk WGBS adult male neurons reference data (*Loyfer et al., 2023*).

CpG sites (N ~116,000), with average methylation between 0.25 and 0.75 in bulk neurons from seven males (*Loyfer et al., 2023*), were further filtered by discarding more stable CpG sites with average methylation less than 0.2 or more than 0.8 for all cells, inhibitory neurons, and excitatory neurons in brain H02. The ~79,000 CpGs were further filtered to remove sites with average methylation less than 0.3 or greater than 0.7 in brain H01, and ~31,000 fCpG sites were used for analysis.

fCpG site methylation appears neutral because they are predominately intergenic, with 16% within genes or promoters (*Supplementary file 1*). Epigenomes from 15,434–21,836 cells were downloaded from three male brains with a general criterion of allc.tsv.gz file sizes 90 mb or larger (*Table 1*). Neurons were preferentially sampled, whereas glial cells were sometimes excluded (*Supplementary file 2*). Analyzed cells had at least 500 fCpGs (average ~1100), with pairwise distances (PWDs) calculated between cell pairs when at least 30 fCpGs were comparable (average ~48 fCpGs per cell pair). Each cell, annotated by its provided phenotype and location, is characterized by its fCpG methylation level and its PWDs from other cells. A PWD of 0 is a perfect match and 0.5 indicates randomization. fCpG methylation was variable between cells with averages of ~58% for all three brains (*Figure 1A*). The 73–197 million possible cell pair comparisons revealed polymorphic barcodes with average PWDs of ~0.47 between cells (*Figure 1B*).

## fCpG barcodes initialize methylated and change with cell division

fCpG barcode patterns were similar between the brains, and data are presented for H01, with H02 and H04 presented in supplemental figures. Methylation was variable between cells of the same type and average methylation was highest in the pons (PN) and thalamus (THM), intermediate for other inhibitory neurons, and lowest for excitatory neurons, glial cells, and cerebellar cells (*Figure 2A*). Outer layer cortical excitatory neurons (L2_3) that are made later during development were less methylated than inner cortical excitatory neurons (L4_6) that appear earlier. Predominantly methylated individual fCpG sites were common in subcortical neurons (PN, THM), less frequent in other inhibitory neurons, and rare in excitatory neurons (*Figure 2B*).

The methylation hierarchy is consistent with a barcode initialized with predominately methylated fCpGs in a brain progenitor cell. Barcodes become progressively demethylated and are fixed when their cells stop dividing and differentiate, which occurs at different times and places during brain development. Barcodes for each cell type had a range of methylation (*Figure 2A*), consistent with synchronous development rather than a strict stepwise process. Very simplistically, the hindbrain with mature neurons (*Fan et al., 2020*) forms early in development, followed by inhibitory neurons in the ganglionic eminences, and then excitatory neurons and glial cells in the cortex. Barcode methylation follows this temporal development and reconstructs when specific neuron types start to appear and reach their adult contents (*Figure 2C*). For example, after barcodes change from ~100 to~70% methylated, most adult brainstem (PN) and inhibitory neurons are present but excitatory neurons are fewer, with very few adult outer layer (L2_3) neurons. Outer excitatory and glial progenitor cells are present (*Eze et al., 2021*), but their barcodes continue to demethylate until they stop dividing and differentiate later in development. This stopwatch-like pattern, with more demethylated barcodes in later appearing cell types, was present in all three adult brains.

## fCpG barcodes are polymorphic

A progenitor cell barcode is assumed to become increasingly polymorphic with subsequent divisions. This pattern was observed, with average barcode PWDs lowest in brainstem cells (PN), intermediate between other inhibitory neurons, and highest for excitatory neurons (*Figure 3A*). Most cells had different barcodes, with an overall average PWD of ~0.47 (*Figure 1B*). Cells of the same phenotype were more similar with lower average PWDs (*Figure 3A and B*), suggesting they are more related to each other and have common progenitors.

The human brain has billions of cells and relatively few cells were sampled from each region. Consistent with sparse sampling, cell pairs with nearly identical barcodes and smaller PWDs (<0.05) were rare. To help distinguish between ancestry and chance, cells within and between brains were compared (*Table 1*). Closely related cell pairs were ~2.9 times more frequent within a brain (average ~9.8 per million) compared to between brains (average ~3.4 per million). Closely related cells had fewer

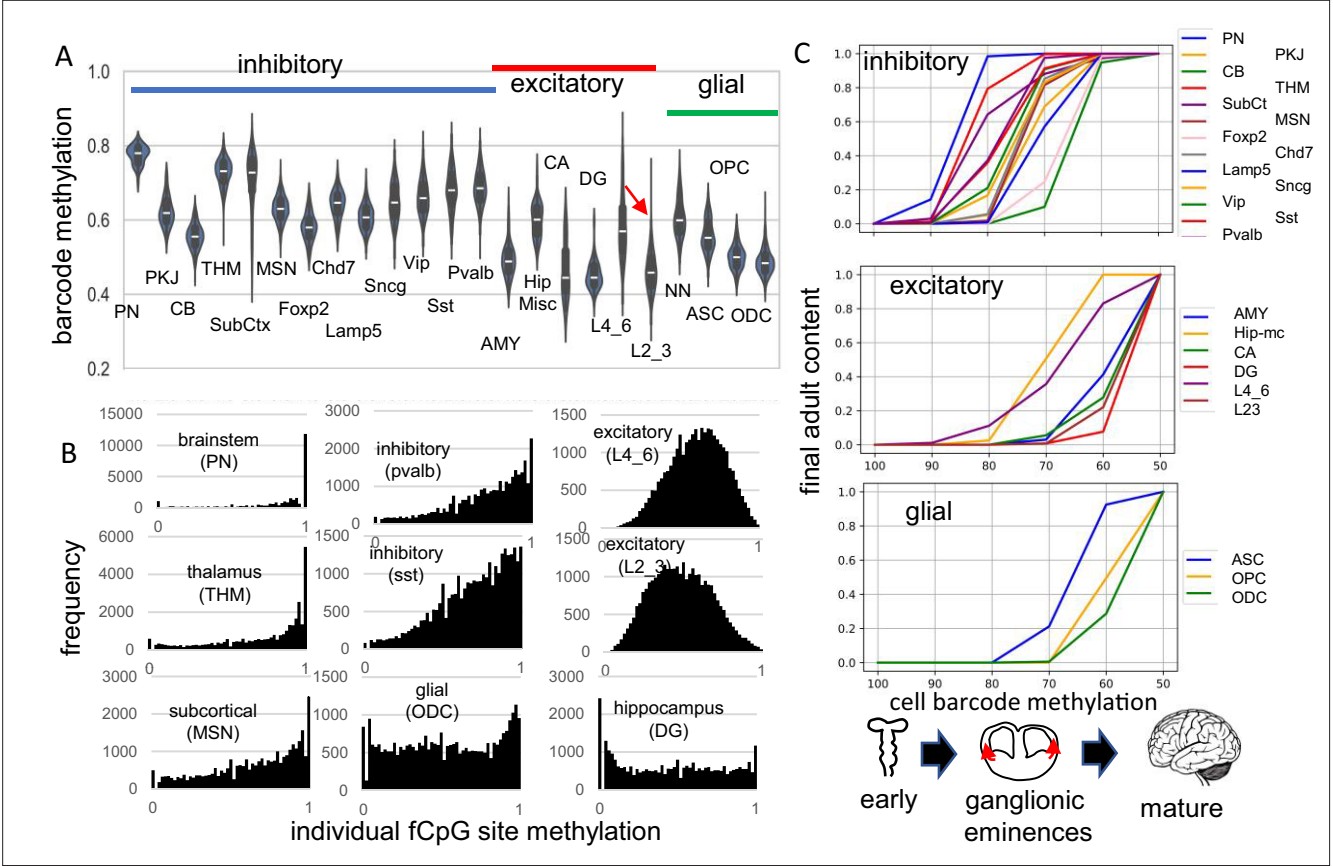

**Figure 2.** Higher fluctuating CpG (fCpG) barcode methylation in earlier emerging cells PN CB SubCt Foxp2 Lamp5 Vip Pvalb excitatory THM MSN Chd7 Sncg Sst AMY Hip-mc CA DG L4_6 L23 glial cell barcode methylation ASC OPC ODC ganglionic eminences mature. (**A**) Average barcode methylation was higher in the brainstem and inhibitory neurons. Barcode methylation was lower for excitatory neurons, cerebellar, and glial cells. Notably, average methylation was lower for outer cortical (L2_3) compared to earlier appearing inner (L4_6) cortical excitatory neurons. Abbreviations are as in reference 3, with L2_3 all outer and L4_6 all inner cortical excitatory neurons, and NN are non-neuronal cells other than ASC, OPC, and ODC. (**B**) Most fCpGs appear to start methylated in a progenitor because nearly all individual fCpGs are methylated in inhibitory neurons in the subcortex (PN, THM, MSN). Many fCpGs in inhibitory neurons (pvalb, sst) are still predominately methylated. Few fCpGs in excitatory neurons that differentiate later in development are fully methylated. Glial cells that also emerge late in development, and hippocampal cells that may divide postnatally had variable methylation with both highly methylated and unmethylated fCpGs. (**C**) Barcodes are assumed to become fixed when their cells stop dividing and differentiate. Therefore, barcode methylation levels can indicate when neurons emerge during development, and can be correlated with a cartoon of physical caudal to rostral brain development. The x-axis indicates the barcode methylation of individual cells and is assumed to roughly correlate with calendar time. The y-axis indicates the cumulative proportion of cells of each type present at each methylation level. A value of 0 indicates that cells of given type are not yet present and a value of 1 indicates the adult content of this cell type has been reached. At the start of development, inhibitory neurons (PN) in the pons with highly methylated barcodes appear first. More inhibitory neurons, made in the ganglionic eminences, appear and reach their final adult contents before many cortical excitatory neurons and glial cells appear. Notably, barcode methylation indicates many lower cortical layer neurons appear earlier in life relative to outer cortical neurons that reach adult levels late in development. Brain contents inferred by adult barcodes may differ from actual neonatal brains because neurons that die during development are not sampled in adult brains.

The online version of this article includes the following figure supplement(s) for figure 2:

**Figure supplement 1.** H02 data.

**Figure supplement 2.** H04 data.

matching fCpG sites (~35 compared to ~48 for all cell pairs) and were more common early in development when barcodes are more methylated (*Figure 3C*), indicating that lower barcode complexity favors matching. Overall, fCpG barcodes are sufficiently polymorphic to distinguish between most adult brain cells.

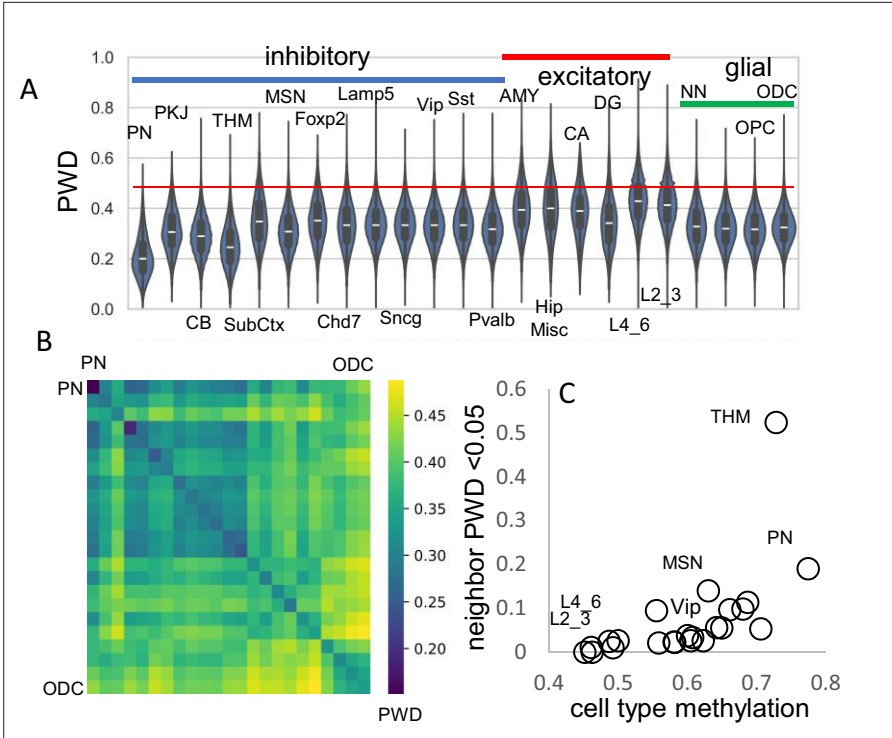

**Figure 3.** Related cell pairs. (**A**) Most cells had different barcodes with average pairwise differences (PWDs) between cell pairs of ~0.5. Cell pairs of the same phenotype had different barcodes but were on average more related to each other. (**B**) Heatmap showing that cells of the same phenotype are more related. (**C**) Cells that emerge early in development are more related and more methylated. Closely related nearest neighbors (PWD <0.05) are numerically more common for more methylated cell types.

The online version of this article includes the following figure supplement(s) for figure 3:

**Figure supplement 1.** H02 data.

**Figure supplement 2.** H04 data.

**Figure supplement 3.** Brain tumor fCpG methylation: fCpG sites were matched with brain tumor data from methylation arrays.

## Brain lineage trees

It should be possible to reconstruct human brain development if barcodes record ancestry. fCpG barcodes from ~1000 brain cells with different phenotypes yield trees with a standard phylogeny software that resemble caudal to rostral development (*Figure 4*). The trees are rooted by a progenitor with a fully methylated barcode, and branches progressively yield brainstem neurons (PN), a subset of excitatory lower (L4_6) neurons, thalamic neurons (THM), inhibitory neurons, cerebellar cells, and glial cells. Excitatory neurons branch last, and hippocampal neurons (CA, DG) that may divide postnatally (*Gage, 2002*) were at the terminus. Cells are generally grouped by phenotype, with some early appearing excitatory neurons admixed among inhibitory neurons. Similar trees were observed for H02 and H04, albeit with less separation between inhibitory and excitatory neurons for H04 (*Figure 4A*). Barcode lineage trees are largely consistent with expected sequential neuronal differentiation.

## Cell lineage fidelity and cortical migration

Uncertain for mouse and human development is whether inhibitory and excitatory neurons originate from shared or distinct progenitors (*Tian et al., 2023*; *Bandler et al., 2022*; *Delgado et al., 2022*). Barcodes could potentially record neuronal differentiation patterns and lineage fidelity can be quantified by comparing most closely related cells or nearest neighbor cell pairs with PWDs <0.05. The approach remains speculative due to several factors: the absence of direct experimental validation, limited experimental cell sampling, and the possibility that barcode similarities may arise by chance rather than reflecting true biological relationships. Lineage fidelity was high (>90%) for inhibitory

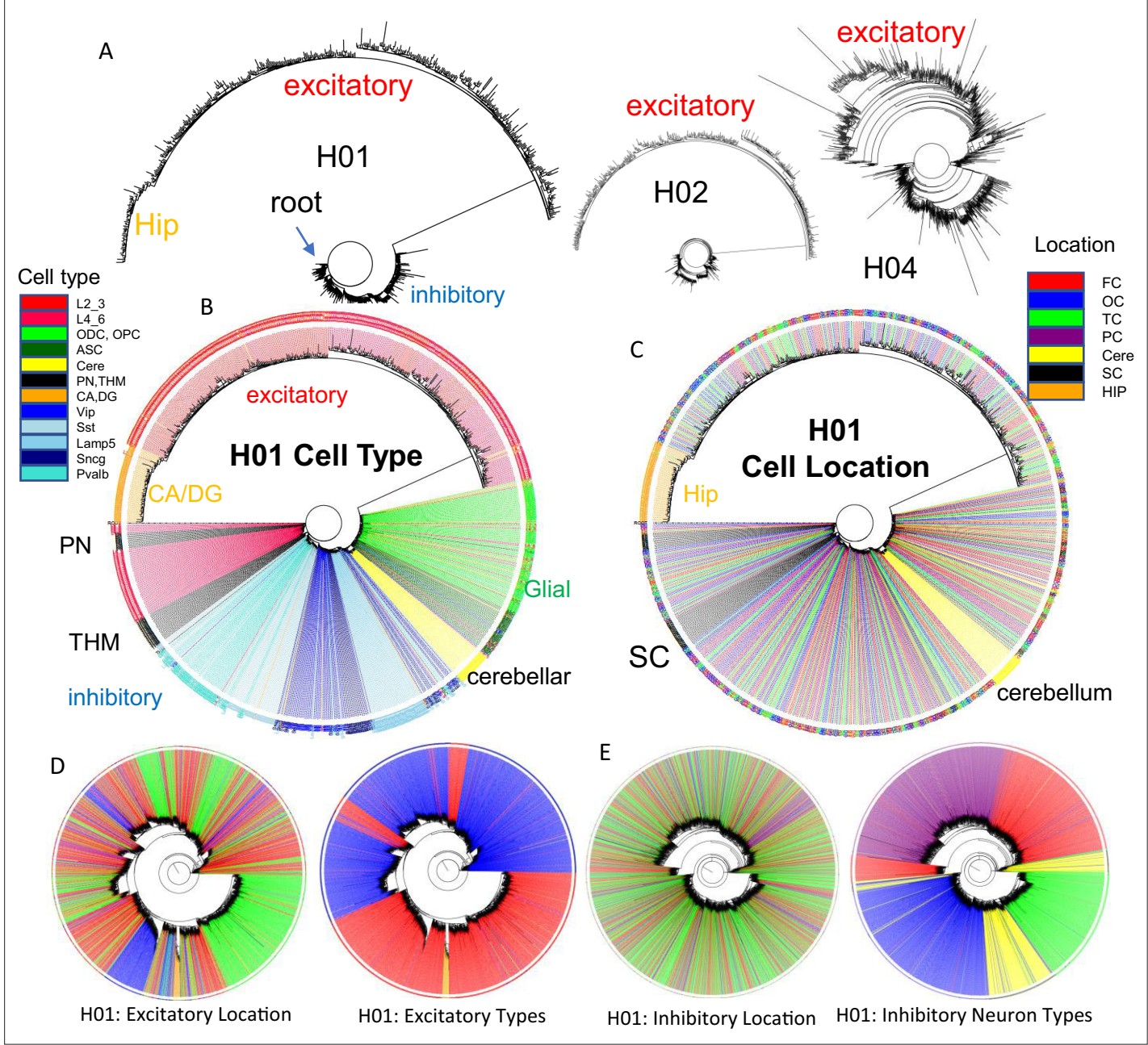

**Figure 4.** Brain trees. (**A**) Barcodes from 960 cells form trees using IQtree (**Nguyen et al., 2015**) that are rooted by a fully methylated progenitor, and generally follow caudal to rostral brain development, with sequential branching of inhibitory neurons, cerebellar neurons, and excitatory neurons, with hippocampal neurons furthest from the start. Trees are similar between the brains, with H04 inferring less distance between inhibitory and excitatory lineages. The trees illustrate the ability to produce phylogenies with IQtree, but the phylogenies are limited by sparse cell sampling and that barcodes may be similar by chance. The degree of confidence was generally low, with bootstrap branch support typically less than 15%. (**B**) H01 tree with labeled cell types. Neuron types generally clustered by phenotypes with closely branching excitatory and inhibitory neurons more common earlier in development. (**C**) H01 tree with labeled cell locations. Related inhibitory and excitatory neurons can be found in different parts of the brain FC = frontal (red), TC = temporal, OC = occipital (blue), PC = parietal, HIP = hippocampus (orange), cere = cerebellum (yellow), SC = subcortical (black). (**D**) H01 tree with ~2853 cortical excitatory neurons has more evidence of localized radial migration because related neurons are more often found in the same cortical region. Excitatory neurons cluster by subtype, and closely related lower and upper excitatory neurons were still few. (**E**) H01 tree with ~2847 cortical inhibitory neurons still retains evidence of tangential migration with related neurons scattered throughout the cortex. Inhibitory neurons cluster by subtype with switching between some closely related pairs.

The online version of this article includes the following figure supplement(s) for figure 4:

**Figure supplement 1.** H02 data.

**Figure supplement 2.** H04 data.

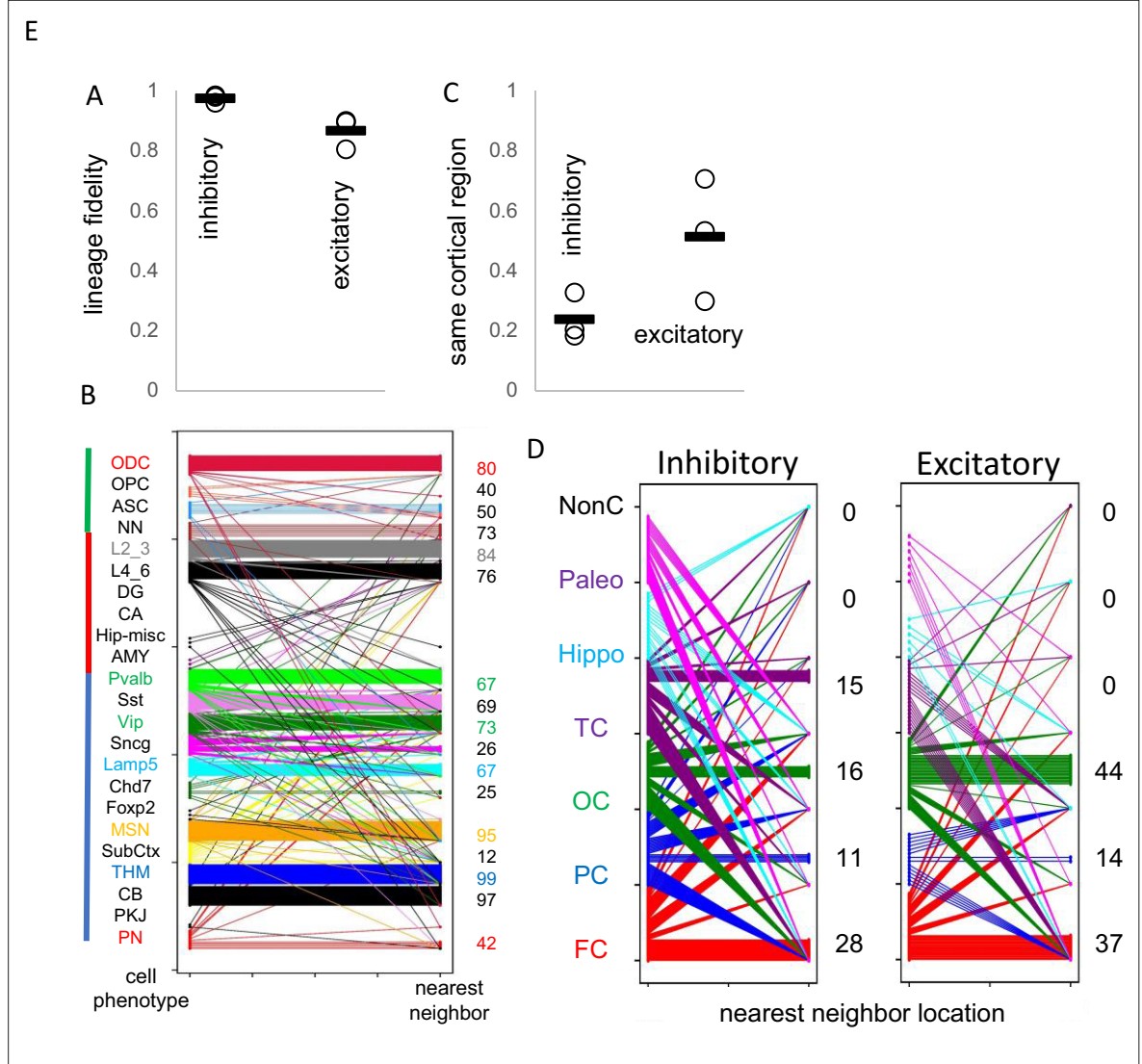

**Figure 5.** Lineage fidelity, migration, and differentiation. (**A**) Inhibitory neurons have higher lineage fidelity because nearest neighbor pairs (pairwise difference, PWD <0.05) were nearly always both inhibitory neurons. Excitatory neurons had slightly less lineage fidelity because a nearest neighbor was more often an inhibitory neuron. Data are for all three brains. (**B**) Nearest neighbor inhibitory neuron pairs often had subtype differences. More lineage fidelity was generally present for subcortical and excitatory neurons. Numbers indicate percent lineage subtype fidelity. (**C**) Nearest neighbor inhibitory and excitatory neuron pairs showed evidence of tangential migration because they were found in different cortical regions. The data indicate greater evidence of inhibitory neuron tangential migration than for excitatory neurons. However, the extent of migration is uncertain because of sparse sampling and because barcodes may be similar by chance. Data are for all three brains. (**D**) Nearest neighbor neurons were scattered in the cortex. Numbers indicate percent location fidelity. (NonC = non-cortical location, Paleo = paleocortex).

The online version of this article includes the following figure supplement(s) for figure 5:

**Figure supplement 1.** H02 data.

**Figure supplement 2.** H04 data.

neurons (*Figure 5A*). Excitatory lineage fidelity was slightly lower, indicating that some excitatory and inhibitory neurons may share common progenitors (*Delgado et al., 2022*). Lineage trees indicate common progenitors are present earlier in development, and excitatory neurons that appear later do not have many closely related inhibitory neighbors (*Figure 4B*). The barcodes documented the known switching between inhibitory neuron subtypes (*Figure 5B*). Hindbrain, excitatory, and non-neuronal cells had more subtype lineage fidelity.

Barcodes can also potentially infer migration because their neurons are annotated by their adult locations. Daughter cells with similar barcodes could be sampled from the same region, or from

different regions if migration occurred. Trees (*Figure 4C*) indicate that most neurons sampled from the brainstem and hippocampal regions are related and localized to their respective regions. Inhibitory neurons were scattered throughout the cortex, consistent with their differentiation in the ganglionic eminences and subsequent tangential migration to the cortex. Nearest neighbor inhibitory cortical neuron pairs were found in the same cortical region ~25% of the time (*Figure 5C and D*). Nearest neighbor excitatory neuron pairs were also scattered throughout the cortex, but less than inhibitory neurons, and were in the same cortical region ~50% of the time. The barcode data indicate more inhibitory rather than excitatory neuron tangential migration, but the extent of migration is uncertain due to sparse sampling and because barcodes can match by chance.

The poor ability to detect localized excitatory neuron radial cortical migration with ~1000 cell whole brain trees (*Figure 4C*) may reflect that sparse sampling is unlikely to include multiple neurons from the same small clonal region that originated from common subventricular progenitors (radial unit hypothesis *Rakic, 1988*). Greater localized excitatory neuron migration was seen when trees were reconstructed with more (~2800) neurons, while inhibitory neurons still showed scattered tangential migration (*Figure 4D*). Neurons of the same subtype were still more related. Hence, lineage trees appear to increase their resolution with more cells, albeit related lower and upper excitatory neuron pairs were still uncommon, which may reflect the unlikely chance of sampling very small radial clonal units.

## Discussion

fCpG barcodes are potential markers of somatic cell ancestry and not cell type classifiers, although cells of the same phenotype are often related because they originate from common progenitors (see new Supplement for fuller discussion). Dynamic barcodes would be useful to study human tissues, but testing their performance is difficult. Ideally, samples obtained at different times would document how they change. The brain facilitates barcode validation because it periodically stores neurons that stop recording at relatively defined times and locations (*McKenna and Gagnon, 2019*; *Finlay and Darlington, 1995*). Specific neuron subsets recovered from the adult brain allow for sampling through time and before birth.

This serial sampling strategy facilitated fCpG barcode validation. The barcode appeared to start predominantly methylated in multiple individuals and became sufficiently polymorphic to distinguish between thousands of neurons. Barcode changes appear to represent replication errors because they reconstruct lineage trees roughly consistent with caudal to rostral brain development. Barcode methylation may indicate when different neurons that survive to adulthood appear in the neonatal brain (*Figure 2C*).

The current barcode indicates that most inhibitory and excitatory neurons have relatively distinct progenitors, consistent with the lineage dendrograms reconstructed with neuron-specific methylation of the same data (*Tian et al., 2023*). There was also evidence for common inhibitory and excitatory progenitors (*Delgado et al., 2022*), primarily for earlier emerging excitatory neurons. Tangential migration was also detected, manifested by inhibitory neurons with closely related barcodes in different cortical regions. Tangential excitatory neuron migration was also detected, albeit related excitatory neurons were more localized than inhibitory neurons. Tangential migration is also seen with sequencing studies that find neurons with specific mutations in multiple brain regions (*Lodato et al., 2015*; *Breuss et al., 2022*; *Chung et al., 2024*; *Evrony et al., 2015*).

fCpGs more efficiently distinguish between cells than mutations due to higher replication error rates. Although average methylation decreases with time, both demethylation and remethylation are likely because fully demethylated neurons were not observed, and balanced fluctuating methylation is inferred in other tissues when CpG sites are ~50% methylated in bulk tissues (*Gabbutt et al., 2022*). More adult divisions in brain cancers did not saturate the barcode with average fCpG methylation ~50% (*Figure 3—figure supplement 3*). Fluctuating methylation complicates lineage tracing but backmutations can be modeled for ancestral reconstructions. Lineage resolution could be improved by combining mutations and fCpGs.

Weaknesses of this study include very sparse cell sampling and lack of uniform CpG site comparisons between neurons. Like many human fate marker studies, it is difficult to independently verify accuracy. Interpretation of the barcodes relies on several untested assumptions, including relatively constant error rates between fCpG sites and through aging, neutrality, and a predominantly fully

methylated start in the progenitor cell. Epigenetic remodeling occurs after progenitors stop dividing (*Ciceri et al., 2024*), which could erase ancestral barcode information. However, neurons of the same type were both closely and remotely related, indicating that such epigenetic remodeling does not systematically alter the fCpG sites. In addition, fCpG barcodes appear to be relatively stable through aging (new Supplement). Inferred lineage trees (*Figure 4*) had relatively low statistical support for their branches and are presented to demonstrate that the barcodes are readily organized into trees with a commonly used phylogeny software.

Technical improvements such as targeted bisulfite sequencing of a limited number of informative fCpGs could lead to more consistent coverage and less expensive sequencing of more neurons. Single-cell measurements of small numbers of fCpGs, and snMCode cell type-specific CpG sites (*Tian et al., 2023*), could efficiently reconstruct human brain lineages, although back changes add complexity. A barcode of 100 fCpGs has enough complexity ($2^{100}$ or $\sim 1 \times 10^{30}$) to potentially distinguish between most excitatory neurons, with less resolution early in development when cells are inherently more related.

The analysis of more brains can verify that a fCpG barcode starts predominately methylated in most individuals. A common initialized state could facilitate standardized human fate maps and comparisons between individuals. Many polymorphisms linked to brain abnormalities such as autism are in neuronal proliferation, migration, and maturation pathways (*Pan et al., 2019*), and this preliminary survey indicates lineage heterogeneity between individuals (*Figure 4A*). fCpG barcodes have been applied to the intestines, endometrium, and blood (*Gabbutt et al., 2022*), and could be found for multiple other tissue types, helping to unravel human development and aging.

## Methods
### Brain single-cells
Single-cells with their annotations and methylation at each fCpG site were read from single-cell files downloaded from GEO (GSE215353) and supplemental files from reference (*Tian et al., 2023*). Lists of fCpG sites and data summarized for the Figures are in *Supplementary file 2*. The cells and methylation at the fCpG sites are in *Supplementary file 3* and *Supplementary file 4*; *Supplementary file 5*. PWDs were calculated between all cell pairs with at least 30 matching fCpG sites, with PWD data matrices in *Supplementary file 6*. Additional details and data in response to the Reviewers are provided in the Appendix.

### Iqtree
IQtree (*Nguyen et al., 2015*) tree was downloaded and run on a server with 64 cpus and 32 GB of memory. The model (GTR2 + FO + G4) accounts for backmutation and binary data with missing values. Bootstraps were 1,000 per tree with 3000 iterations for whole brain (~1000 cells) trees and 1000 iterations for inhibitory or excitatory (~2,800 neurons) trees. Trees (.treefile) were displayed with FigTree (http://tree.bio.ed.ac.uk/software/figtree/) with truncation of long branches (generally fewer than 10) for display purposes. The cells used for the trees are in *Supplementary file 6*.

## Acknowledgements
This work was supported by grants from the NIH (P01CA196569 and CA271237). I thank Drs. Trevor Graham and Heather Grant for useful discussions, and Omar Khan and Nikhil Krishnan for initial studies. The author thanks all of the researchers that helped produce the high-quality very valuable, and freely available data used for analysis.

## Additional information

### Funding

| Funder | Grant reference number | Author |
|--------|------------------------|--------|
| National Institutes of Health | P01CA196569 | Darryl Shibata |
| National Institutes of Health | CA271237 | Darryl Shibata |

The funders had no role in study design, data collection and interpretation, or the decision to submit the work for publication.

### Author contributions
Darryl Shibata, Conceptualization, Data curation, Software, Formal analysis, Funding acquisition, Validation, Visualization, Methodology, Writing – original draft, Writing – review and editing

### Author ORCIDs
Darryl Shibata (iD) https://orcid.org/0000-0002-4567-1639

Reviewer #1 (Public review): https://doi.org/10.7554/eLife.101163.3.sa1
Reviewer #3 (Public review): https://doi.org/10.7554/eLife.101163.3.sa2
Author response https://doi.org/10.7554/eLife.101163.3.sa3

## Additional files

### Supplementary files
MDAR checklist

Supplementary file 1. fCpG annotations.

Supplementary file 2. fCpG genomic locations.

Supplementary file 3. fCpG Brain H01 data.

Supplementary file 4. fCpG H02 brain data.

Supplementary file 5. fCpG H04 brain data.

Supplementary file 6. Cells used for IQtree analysis.

Source code 1. Python code used for analysis.

### Data availability
The data used were obtained from GEO (GSE215353).

The following previously published dataset was used:

| Author(s) | Year | Dataset title | Dataset URL | Database and Identifier |
|-----------|------|---------------|-------------|-------------------------|
| Tian W, Bartlett A, Liu H, Altshul J, Nery JR, Chen H, Ecker JR, Gomez CR | 2023 | Epigenetic landscape of Human Brain by Single Nucleus Methylation Sequencing | https://www.ncbi.xyz/geo/query/acc.cgi?acc=GSE215353 | NCBI Gene Expression Omnibus, GSE215353 |

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

## Appendix 1

The purpose of this appendix is to address the broad concerns raised by the three reviewers. The goals are to better explain the mechanics of fCpG barcodes, present new data illustrating their reproducibility and stability with aging, and demonstrate how barcodes can record ancestry after conception, even in the context of active demethylation.

### Differences between fCpGs and traditional cell type classifiers

Both fCpG barcodes and traditional DNA methylation-based cell type classifiers can differentiate between cells, but their identification processes and underlying biological principles differ significantly. Traditional cell type classifiers focus on identifying CpG sites that are consistently differentially methylated across distinct cell types. Success in this approach yields a barcode, such as the snMCode CpG sites (*Tian et al., 2023*), where each cell of a given phenotype shares the same methylation pattern, distinct from those of other phenotypes.

In contrast, the identification of fCpGs is fundamentally different because fCpG methylation can vary among cells, even within the same type. In traditional screening for cell type classifiers, a CpG site is retained if its methylation is consistent across cells with the same phenotype and discarded if it varies. Conversely, when screening for fCpG sites, a CpG site is retained if its methylation differs among cells and discarded if it skews toward either methylated or unmethylated states. The fCpG barcode captures random errors accumulated throughout development, recording ancestry, while the methylation profiles used in cell type classifiers more accurately reflect terminal differentiation.

### Lineage tracing: Cladistics and ancestry

Although fCpG selection and cell type classifier selection are fundamentally opposed, their barcodes may align if clades of cells sharing the same phenotypes arise from common progenitors (*Appendix 1—figure 1A*). This study supports the notion that neurons of the same type typically originate from common progenitors, as cells within a clade generally exhibit polymorphic yet more similar fCpG barcodes than those from different clades. fCpG barcodes may provide complementary insights into cell mitotic ages, diversity within a clade, and migration patterns of daughter cells—factors that are often more challenging to discern from RNA-seq data alone.

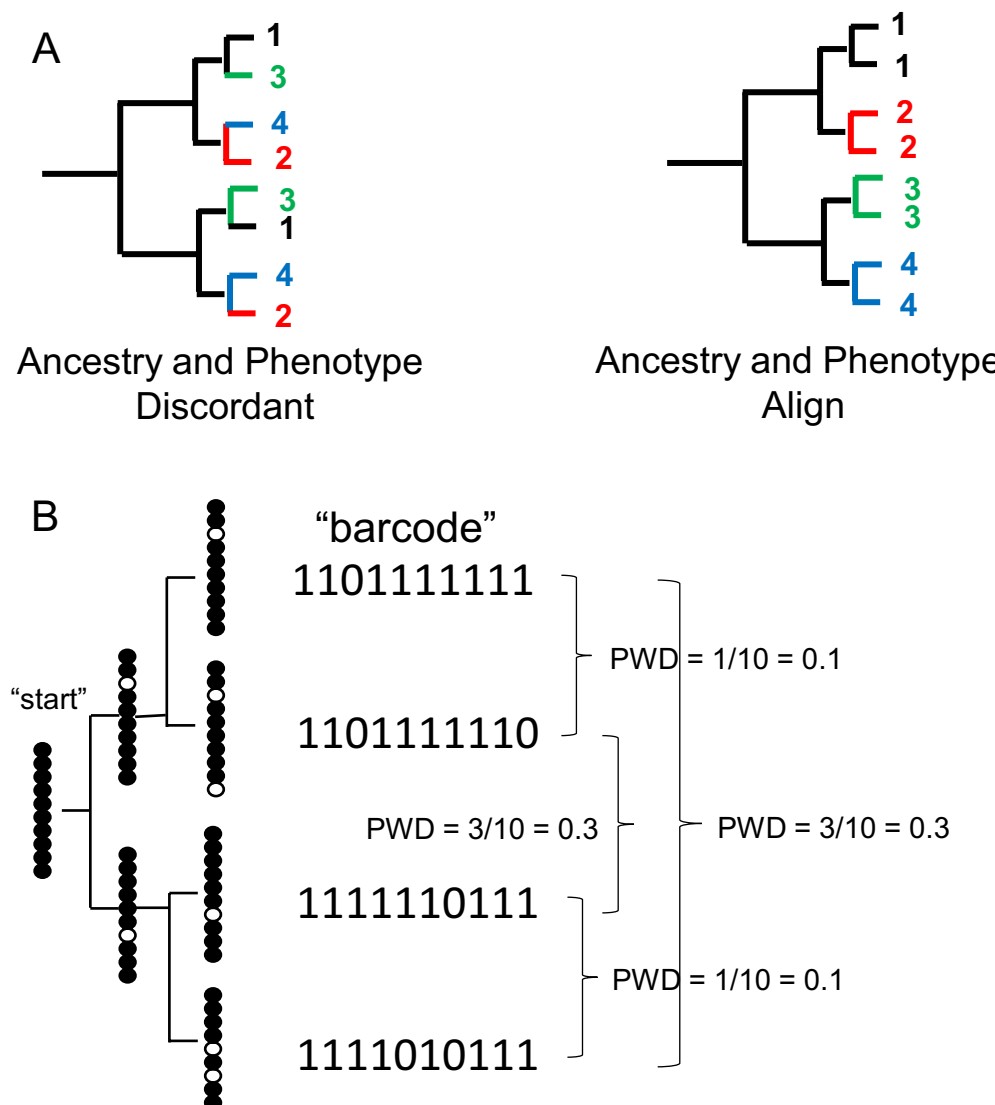

**Appendix 1—figure 1.** Single cell lineage tracing with dynamic fCpG barcodes. (**A**) Trees can be reconstructed by comparing phenotypes or by comparing genomic differences such as fluctuating CpG (fCpG) barcodes. Ancestry and phenotypes may be discordant if progenitor cells produce cells of different phenotypes. More typically, ancestry and phenotypes align because cells with the same phenotypes tend to have common progenitors. For the single-cell brain data, ancestry and phenotype align because cells of the same type are generally more closely related. (**B**) fCpG barcodes appear to start predominately methylated in the progenitor cell. With division, random replication error occur and are propagated to daughter cells. Counting and then averaging the differences between fCpG sites yields an average pairwise distance (pairwise difference, PWD, range 0–1). More related daughter cells tend to have lower PWDs, but barcodes may also match by chance.

## fCpG barcode mechanics

The fCpG barcode begins as predominately methylated, reflecting the observation that inhibitory neurons in the pons—emerging early in development—are predominantly methylated across the three brains examined. It is assumed that random errors occur when the barcode is replicated during cell divisions, as illustrated in a cartoon (*Appendix 1—figure 1B*). As errors are perpetuated and new ones accumulate, closely related cells tend to exhibit more similar methylation patterns than those that are less related. The differences among these patterns can be quantified by calculating the average PWDs between barcode fCpGs.

The barcode mechanics are more formally described by simulations that match the experimental data. These simulations start with a single-cell possessing a fully methylated barcode. Key parameters

include fCpG methylation error rates, the number of cell divisions, and whether a cell division yields two, one, or zero daughter cells. A simulation that broadly matches the experimental data for excitatory neurons has 150 cell divisions and a fCpG error rate of 0.01 per division, applied equally to both methylated-to-unmethylated and unmethylated-to-methylation flips (*Appendix 1—figure 2*).

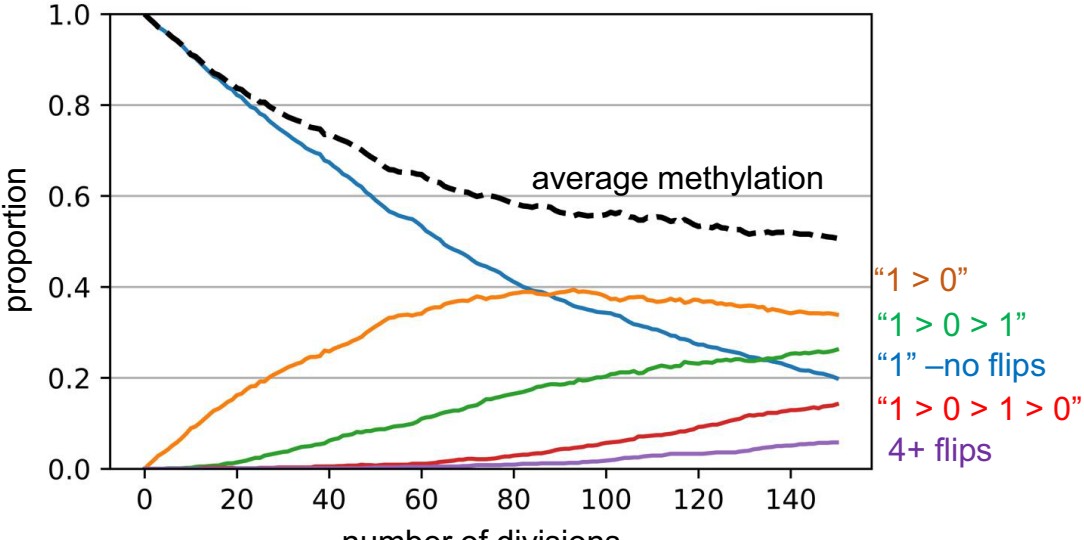

**Appendix 1—figure 2.** Barcode dynamics. Simulations broadly consistent with the experimental data indicate a replication error rate of 0.01 per fluctuating CpG (fCpG) site per division, with equal probabilities of changes or flips from methylated to demethylated (1>0) and from 0>1. A simulation for excitatory neurogenesis is shown, where simplistically, excitatory neurons cease division and appear after 150 divisions. The graph displays how individual fCpG sites change through time. The fCpG barcode starts methylated, and barcode methylation decreases with divisions. Even with an error rate of 0.01, after 150 divisions only about 5% of fCpG sites experience four or more flips, and half have had zero or only a single flip. A fCpG barcode can still effectively distinguish between cells if the flips are random and multiple fCpG sites are compared between cells. Although backflips complicate analysis, the large numbers of replication errors facilitate comparisons between neurons that develop during a short prenatal interval.

Notably, even with a high error rate of 0.01 per division ancestry may be reconstructed because each fCpG sites experiences relatively few changes or methylation flips. By the end of 150 divisions, the average methylation level approaches approximately 50%, with most fCpG sites experiencing three or fewer changes in methylation. Among the methylated sites, around 20% remain unchanged (methylated), while approximately 25% of fCpG sites have undergone two flips (from 1 to 0 to 1). For unmethylated sites, about one-third experience one flip (from 1 to 0), and roughly 15% have had three flips (from 1 to 0 to 1–0). Only about 5% of sites experience more than three flips. Despite this, the randomness of these flips leads to highly polymorphic barcodes among the final cells, with more related cells having more similar barcodes. As noted by Reviewer 1, this barcode mechanism is essentially a mitotic clock that becomes polymorphic.

## Simulations of neurogenesis

Simulations that align with experimental data can effectively illustrate the mechanics of the barcode. The first simulation, designed to model early neurogenesis in the hindbrain, features a limited number of divisions and a basic exponential expansion, beginning with a single-cell possessing a fully methylated barcode (*Appendix 1—figure 3A*). After 19 divisions, resulting in ~500,000 cells, the average methylation across the population declines to approximately 85%, while the average PWD between cells increases to about 0.28. The proportion of closely related neighboring cells (with PWDs <0.1) progressively decreases to around 45%.

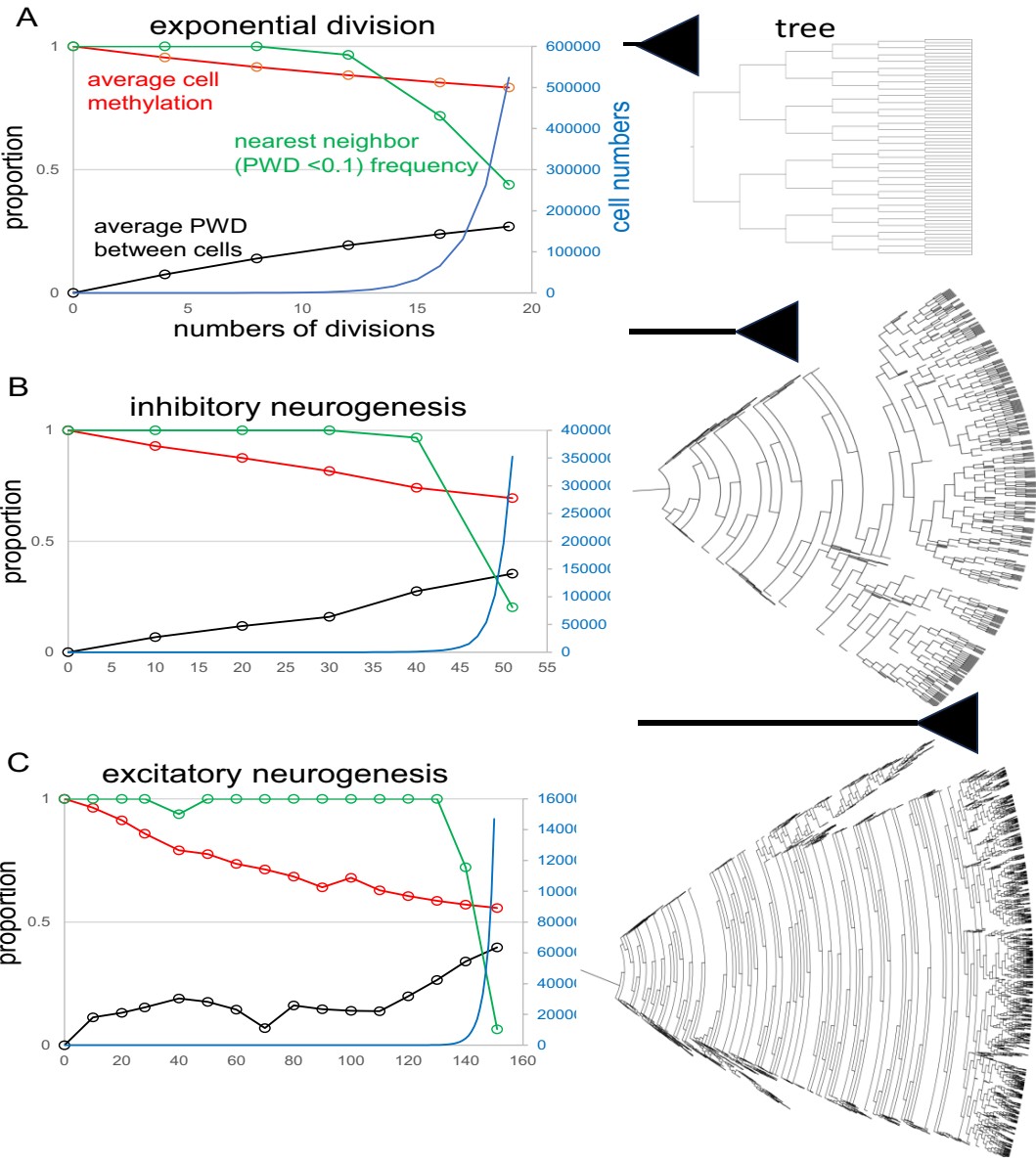

**Appendix 1—figure 3.** Fluctuating CpG (fCpG) barcode simulations. The simulations start with a single progenitor and a fully methylated barcode with 200 fCpG sites and an error rate of 0.01 per division. At each time point, up to 1000 cells are sampled from the population to calculate fCpG barcode average methylation, average pairwise difference (PWD), and cell proportions with a nearest neighbor with a PWD <0.1. Final tree expansions are truncated to allow for visualization. A range of simulations are broadly consistent with the experimental data. (**A**) Simulation of exponential growth where each cell yields two daughter cells broadly models early hindbrain neurogenesis. After 19 divisions, average barcode methylation is ~0.83, PWD is ~0.27, and among the 1000 sampled cells, for about 44% of cells there is another cell with a similar barcode (PWD <0.1). (**B**) Simulation of inhibitory neurogenesis with differentiation after 50 divisions. Early divisions are characterized by cell death (zero or one daughter, represented by dead ends in the tree), with terminal growth. After 50 divisions, average barcode methylation is ~0.69, average PWD is 0.35, and 20% of sampled cells have a nearest neighbor (PWD <0.1) C: Simulations of excitatory neurogenesis with differentiation after 150 divisions. As with inhibitory neurogenesis, cell death during early divisions limits population size before terminal expansion. After 150 divisions, average methylation is ~0.56, PWD is ~0.4, and ~6% of cells have a nearest neighbor (PWD <0.1) among the 1000 sampled cells.

The simulation of inhibitory neurogenesis extends to 50 divisions, also starting from a single-cell with a fully methylated barcode (*Appendix 1—figure 3B*). In this scenario, cell division is balanced by

cell death to maintain a small population early in development, followed by subsequent expansion. By the end of 50 divisions, average cell population methylation falls to about 70%, and average PWD increases to around 0.35, with the proportion of closely related neighboring cells dropping to approximately 20%.

The simulation of excitatory neurogenesis involves 150 divisions, where early development similarly balances cell division and cell death, followed by expansion (*Appendix 1—figure 3C*). The average cell population methylation decreases to about 55%, and the average PWD among cells rises to around 0.4, with the proportion of closely related neighboring cells declining to about 5%. While these simulations do not fully capture the complexity of neurogenesis, such as the variation in differentiation times, they illustrate how fundamental barcode mechanics can yield neuron populations broadly consistent with experimental data.

### Barcode cell division dynamics during early development

The public review requested additional experimental data, and Reviewer 1 raised concerns that active methylation processes early in development could obscure any signature of inaccurate fCpG methylation. Direct comparisons of immediate human daughter neurons pose challenges, and it is uncertain what a relevant cell culture study might be.

Whole genome bisulfite single-cell data are available for human germ cells, zygotes, 2–8 cell embryos, morulae, and ICM samples (*Li et al., 2018*). The data show that barcodes are more methylated in germ cells and gradually become less methylated as development progresses to the ICM (*Figure 4*). While fCpG barcodes are polymorphic between unrelated samples, they exhibit greater similarity among closely related cells in the 2–8 cell stage, with reduced similarity in morulae and ICM samples (*Appendix 1—figure 4*).

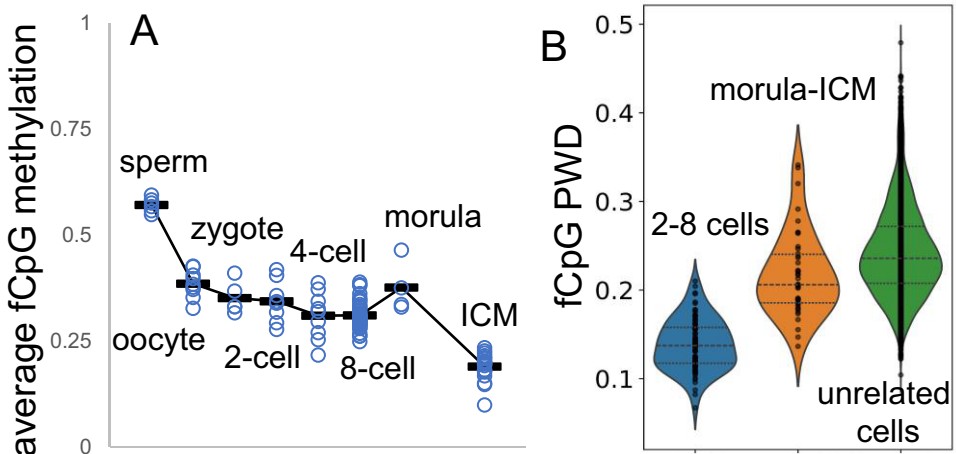

**Appendix 1—figure 4.** Fluctuating CpG (fCpG) barcodes at conception, when germline methylation is erased by active and passive demethylation. Whole genome bisulfite single-cell sequencing data are from GSE100272. Male cells were inferred from a paucity of Y chromosome reads. (**A**) fCpG methylation generally decreases during early development. fCpG barcode methylation is highest in sperm, albeit sperm X chromosomes yield female zygotes. A brain cell progenitor with predominately methylated fCpGs was not evident. (**B**) Unlike at the start of brain development, fCpG barcodes early in life are polymorphic between unrelated embryos. However, fCpG barcodes are more similar between related cells in 2–8 cell embryos, and less similar between cells in morulae and the ICM. Dots indicate values of cell pairs, with a minimum of 25 comparable fCpG sites.

This additional data indicates that more closely related daughter cells have more similar barcodes, even amidst the chaotic background of early development, where both active and passive demethylation largely erases germ cell methylation (*Messerschmidt et al., 2014*). Notably, although a neurogenic lineage is present among the embryonic cells, none of the cells displayed fully methylated barcodes, suggesting that the anticipated remethylation in a neurogenic progenitor

has not yet occurred. The activation of DNMT3A during early neurogenesis (*Lister et al., 2013*) may ultimately lead to the establishment of fully methylated barcodes that subsequently become polymorphic.

## Neuron barcode reproducibility and stability with aging

New single-cell neuron datasets (*Heffel et al., 2024*; *Chien et al., 2024*) became available while the manuscript was under review. The core concept behind brain ancestry barcodes is that they can be used to compare different brains, assuming all brains start with a fully methylated barcode and share similar developmental trajectories. These new datasets offer valuable opportunities to test for barcode reproducibility and stability. One dataset (*Appendix 1—figure 5*) measured neurons in a 7-mo-old brain from the frontal cortex and hippocampus (*Heffel et al., 2024*). The adult fCpG barcode methylation levels observed are largely present in the infant brain, supporting the idea that fCpG methylation patterns are established prenatally through replication errors and remain stable once cell division ceases.

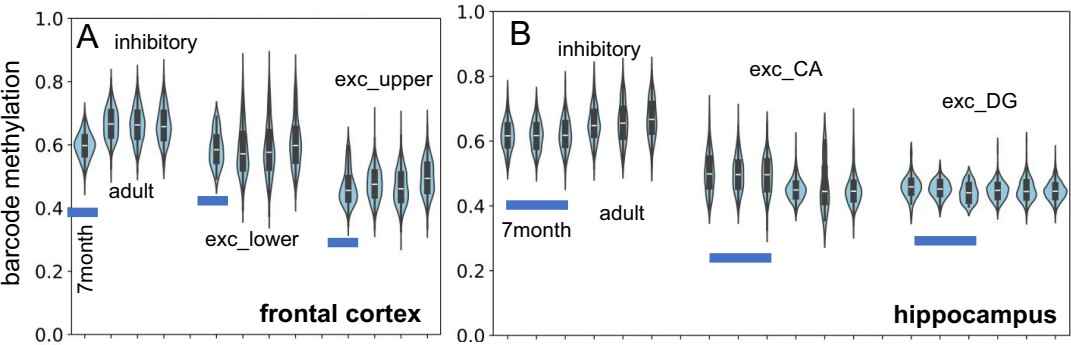

**Appendix 1—figure 5.** New single-cell data indicate fluctuating CpG (fCpG) barcode methylation at 7 mo of age is similar to adult levels (H02, 29 yo; H01, 42 yo; H04, 58 yo). (**A**) Inhibitory, and lower and upper cortical excitatory neuron barcode methylation levels from the frontal cortex are similar. (**B**) Inhibitory and excitatory (CA and DG) neuron barcode methylation levels are similar between infant (three samples) and adult hippocampus.

A second dataset (*Chien et al., 2024*) sampled the same region of the frontal cortex (Brodmann area 46) from young (~25 y) and old (~70 y) brains. This study did not reveal extensive changes in DNA methylation with aging but identified a small number of differentially methylated CpG regions (*Chien et al., 2024*). Consistent with the overall stability of DNA methylation during aging, average barcode methylation (*Appendix 1—figure 6A*) and PWDs (*Appendix 1—figure 6B*) among the ten samples from six male brains and three adult brains (Brodmann area 46) in the manuscript were similar, indicating reproducibility. While average barcode methylation remained stable with age for inhibitory neurons, upper and lower cortical excitatory neurons exhibited significantly higher methylation levels in older brains. This trend suggests a preferential loss of barcode methylation in 'older' excitatory neurons, as significantly fewer neurons with less methylated barcodes were sampled from older brains (*Appendix 1—figure 6C and D*). Alternatively, if fCpG methylation remains stable after cell division ceases, the observed decrease in average barcode methylation could indicate that neurons with less methylated barcodes are preferentially lost with aging. This suggests that neurons arising later in development may have higher mortality during aging. While other explanations are possible, the data illustrate how barcodes could be potentially used to infer both brain development and aging. Overall, these two new datasets help illustrate the degree of fCpG barcode reproducibility, and stability during aging.

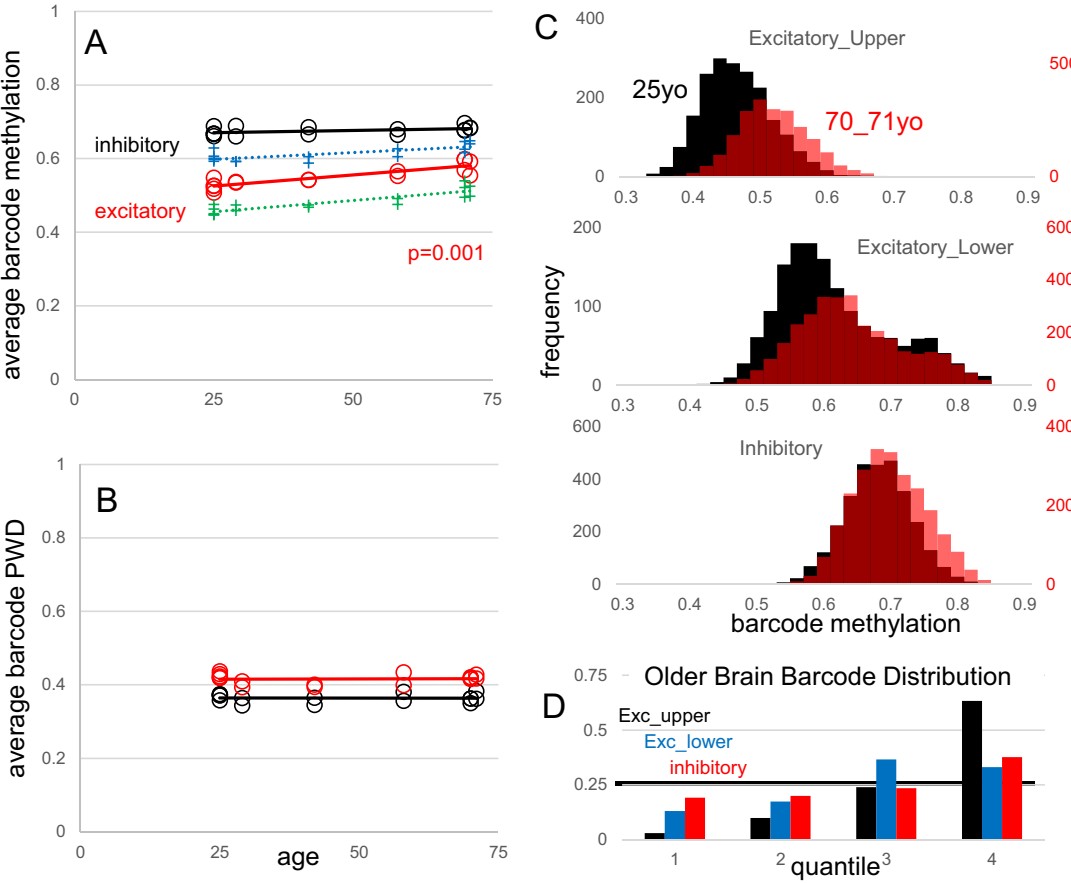

**Appendix 1—figure 6.** Fluctuating CpG (fCpG) barcode reproducibility and stability with aging. New data are frontal cortex (Broadman area 46) WGBS single-cells from young (25 y-old, three individuals and five samples) and older (70–71 y-old, three individuals and fivesamples) males. Data from the manuscript are also shown (H02, 29 yo; H01, 42 yo; H04, 58 yo). (**A**) Average neuron barcode methylation levels were similar between different aged individuals. Excitatory neuron barcode methylation in older males (70–71 yo) was significantly greater than the 25 yo males (t-test comparing only the 25 yo and 70–71 yo groups). Both upper (green) and lower (blue) cortical excitatory neurons showed greater average barcode methylation with aging. (**B**) Average inhibitory and excitatory neuron barcode pairwise differences (PWDs) (comparing within subtypes in each brain) were similar between different aged individuals, indicating that barcodes remain polymorphic. (**C**) Composite histograms of individual neuron barcode methylation levels for younger (black, 25 yo) and older (red, 70-71 yo) brains. There is a preferential loss of neurons with less methylated barcodes, especially with excitatory neurons. (**D**) Younger brain neuron barcodes were used to define quantiles. Older brain neurons with less methylated barcodes were depleted in the less methylated quantiles, with significant differences for all quantiles (Mann Whitney U test, p<10–9).

## Summary

This supplement addresses several concerns raised during the Public Review. The criteria for the fCpG barcode, as stated in the manuscript, are as follows: (1) a defined initial pattern in a progenitor cell; (2) polymorphic changes upon cell division; (3) sufficient polymorphism to distinguish between most cells; and (4) the capability to record ancestry. Among these criteria, only the adequacy of barcode polymorphisms to differentiate between most inhibitory and excitatory neurons has been well-established, indicating that while evidence for lineage tracing is supportive, it remains inadequate and dependent on assumptions that are difficult to validate.

The supplement provides the requested more detailed explanation of barcode mechanics. Simulations that begin with a single-cell containing a fully methylated barcode broadly align with experimental data, incorporating relatively straightforward cell divisions, deaths, and expansions alongside random barcode replication errors. The brain facilitates serial barcode examinations because neurons typically stop dividing at different times during development. New barcode data reveal that daughter cells exhibit greater relatedness during embryogenesis, even amidst active and

passive demethylation. Additionally, new data demonstrate the degree of barcode reproducibility and stability throughout aging.

Tools to reconstruct human brain development are limited, and fCpG barcodes can potentially add complementary information such as mitotic ages, diversity within a clade, and migration patterns between immediate daughter cells. fCpG barcode complexity is required because most brain development occurs prenatally, necessitating very high replication error rates to distinguish between billions of adult cells. Such high error rates ($\sim 10^{-2}$) would inherently lead to polymorphic methylation patterns, complicating the differentiation of ancestral signals from technical or biological noise, especially if methylation remodeling occurs independently of cell division. While the current evidence for fCpG barcode lineage tracing is inadequate, a functional DNA methylation-based barcode comprising as few as 50 fCpG sites (yielding $10^{15}$ unique binary patterns) with similar high replication error rates could feasibly reconstruct aspects of human brain development and aging.

## Pipelines

(Files are in Pileline.zip)

1: Download GSE file from GEO

2: Filter out smaller files (typically keep 70 mB or larger tsv.gz files)

3: Use Program 1 ('mergetsvgzcol123divide4by5.py' with 'locALLforfirstfilter.csv') to select CpG sites at the chromosome locations (+and - strands) of the ~31 k fCpGs, and calculate methylation. A.tsv.gz file produces a.tsv file.

4: Use Program 2 ('collectmasterlistfromsmalltsvfastermerge' with 'master_list.csv') to make a csv file with the methylation of each fCpG site for each cell, with blank sites labeled as 'NA'

5: Use Excel to annotate single-cells with their phenotypes from the published keys (1,5). Also calculate the fCpG methylation, and number of fCpGs with data for each cell.

6: Filter cells without relevant phenotypes, and cells with insufficient numbers of fCpG sites with data (500 for the manuscript and 400 fCpG sites for the new Supplement).

7: Calculate PWDs and the number of fCpGs that are compared between all possible cell pairs. Program 3 ('pwd2.py') compares between all cells in a single.csv file (no headers, with individual cells across and fCpGs down). Program 4 ('pwdsquare.py') calculates PWDs and the number of fCpGs compared between cells when two.csv files are compared. Programs 3 and 4 were used to segment the data into smaller chunks to shorten run times. Each program produces a csv file with PWDs and a csv file with numbers of fCpGs compared.

8: Excel was used to remove cell pairs with too few comparable fCpGs (minimum of 30 for the manuscript and 25 for the new Supplement).

9: Excel was used to calculate average PWDs between all comparable cell pairs and to find the minimum PWD of its nearest neighbor.

## Simulations of neurogenesis

Program 5 ('NumPydet_lin_fCpG_5_1.3arrayflipavemultrun_oneout_good_writeN_XeditDS.py') is used to simulate different growth scenarios and is parameterized with two csv files. The first csv file ('lookup5_table_Xonly.csv' with four columns; lineage, array1, p0_1, p1_0) initializes the first progenitor cell. The number of rows is the number of fCpG sites, array1 is the starting methylation state (all 1's), p0_1 is the probability of flipping from 0 to 1, and p1_0 is the probability of flipping from 1 to 0 and is 0.01 in the simulations. The other csv file ('cellnumber_table.csv' with five columns; division, desired_population, q, r, s) controls cell population size and cell death with each division. The desired_population provides the number of cells after each cell division. The program determines cell survival by randomly selecting (without replacement) for each mother cell a q, r, or s value, where (q+r + s) = (the number of mother cells). A cell with 'q' will have one surviving daughter cells, 'r' has 2 surviving daughter cells and 's' has no surviving daughter cells.

The outputs are 'sumlineage_arrays.csv' that calculates the average methylation of the cell population at each fCpG site, and 'last_run_lineage_arrays.csv' which outputs the methylation of each fCpG site for each simulated cell. Program 6 ('covertrowlosecommas.py') produces 'out.csv' to format the data for Program 7 ('pwdofNcellsNaN4sigAVE_Rruns.py') that samples specific numbers of simulated cells to match the numbers of cells sampled with the experimental data. Program 8 ( flip101010.py) was used for the simulations of *Appendix 1—figure 2*.

