## [Editor Report · eLife Assessment]

This study presents a **valuable** conceptual approach that cell lineage can be determined using methylation data. However, the evidence supporting the claims of the author remains **incomplete** after revision. If clarified further as described in the reviews, this approach could be of broad interest to neuroscientists and developmental biologists.

---

## [Referee Report · Reviewer #1 (Public review)]

Summary:

In this manuscript, Shibata describes a method to assess rapidly fluctuating CpG sites (fCpGs) from single-cell methylation sequencing (sc-MeSeq) data. Assuming that fCpGs are largely consistent over time with changes induced by inheritable events during replication, the author infers lineage relationships in available brain-derived sc-MeSeq. Supplementing current lineage tracing through genomic and mitochondrial mosaic variants is an interesting concept that could supplement current work or allow additional lineage analysis in existing data.

However, the author failed to convincingly show the power of fCpG analysis to determine lineages in the human brain. While the correlation with cellular division and distinction of cell types appears plausible and strong, the application to detect specific lineages is less convincing. Aspects of this might be due to a lack of clarity in presentation and erroneous use of developmental concepts. However, without addressing these problems it is challenging for a reader to come to the same conclusions as the author.

On the flip side, this novel application of fCpGs will allow the re-use of existing sc-MeSeq to infer additional features that were previously unavailable, once the biological relevance has been further elucidated.

Strengths:

• Novel re-analysis application of methylation data to infer the status of fCpGs and the use as a lineage marker

• Application of this method to an innovative existing data set to benchmark this framework against existing developmental knowledge

Weaknesses:

• Inconsistent or erroneous use of neurodevelopmental concepts which hinders appropriate interpretation of the results.

• Somewhat confusing presentation at times which makes it hard to judge the value of this novel approach.

---

## [Referee Report · Reviewer #3 (Public review)]

Summary:

Cell lineage tracing necessitates continuous visible tracking or permanent molecular markers that daughter cells inherit from their progenitors. To successfully trace cell lineages, it is essential to generate and detect sufficient new markers during each cell division. Thus, molecular cell lineages have been predominantly studied with stably inherited genetic markers in animal models and somatic DNA mutations in the human brain. DNA methylation is unstable across cell divisions and differentiation, and is hardly called barcodes. The use of "Human Brain Barcodes" in the title and across the whole paper lacks convincing evidence - it is questionable that CpG methylation is always stably inherited by daughter cells.

Strengths:

Analysis of DNA methylation.

Weaknesses:

The unstable nature of CpG methylation would introduce significant problems in inferring the true cell lineage. To establish DNA methylation as a means for lineage tracing, it is necessary to test whether the DNA methylation patterns can faithfully track cell lineages with in vitro differentiated & visibly tracked cell lineages.

The unreliable CpG methylation status also raises the question of what the "Barcodes" refer to in the title and across this study. Barcodes should be stable in principle and not dynamic across cell generations, as defined in the Reference #1. The CRISPR/Cas9 mutable barcodes or the somatic mutations may be considered barcodes, but the reviewer is not convinced that the "dynamic" CpG methylation fits the "barcodes" terminology. This problem is even more concerning in the last section of the results, where CpG status fluctuates in post-mitotic cells.

The manuscript frequently states assumptions in a tone of conclusions and interprets results without rejecting alternative hypotheses. For example, the title "Human Brain Barcodes" should be backed with solid supporting evidence. For another example, the author assumed that the early-formed brain stem would resemble progenitors better and have a higher average methylation level than the forebrain - however, this difference in DNA methylation status could well reflect cell-type-specific gene expression instead of cell lineage progression.

Other points:

(1) The conclusion that excitatory neurons undergo tangential migration is unclear - how far away did the author mean for the tangential direction? Lateral dispersion is known, but it is hard to believe that the excitatory neurons travel across different brain regions. More importantly, how would the author interpret shared or divergent methylation for the same cell type across different brain regions?

(2) The sparsity and resolution of the single-cell DNA methylation data. The methylation status is detected in only a small fraction (~500/31,000 = 1.6%) of fCpGs per cell, with only 48 common sites identified between cell pairs. Given that the human genome contains over 28 million CpG sites, it is important to evaluate whether these fCpGs are truly representative.

(3) While focusing on the X-chromosome may simplify the identification of polymorphic fCpGs, the confidence in determining its methylation status (0 or 1) is questionable when a CpG site is covered by only one read.

---

## [Author Response]

The following is the authors’ response to the original reviews.

**eLife assessment:**
Developing a reliable method to record ancestry and distinguish between human somatic cells presents significant challenges. I fully acknowledge that my current evidence supporting the claim of lineage tracing with fCpG barcodes is inadequate. I agree with Reviewer 1 that fCpG barcodes are essentially a cellular division clock that diverges over time. A division clock could potentially document when cells cease to divide during development, with immediate daughter cells likely exhibiting more similar barcodes than those that are less related. Although it remains uncertain whether the current fCpG barcodes capture useful biological information, refinement of this type of tool could complement other approaches that reconstruct human brain function, development, and aging.

Due to my lack of clarity, the fCpG barcode was perceived to be a new type of cell classifier. However, it is fundamentally different. fCpG sites are selected based on their differences between cells of the same type, while traditional cell classifiers focus on sites with consistent methylation patterns in cells of the same type. Despite these opposing criteria, fCpG barcodes and traditional cell classifiers may align because neuron subtypes often share common progenitors. As a result, cells of the same phenotype are also closely related by ancestry, and ex post facto, have similar fCpG barcodes. fCpG barcodes are complementary to cell type classifiers, and potentially provide insights into aspects such as mitotic ages, diversity within a clade, and migration of immediate daughters---information which is otherwise difficult to obtain. The title has been modified to “Human Brain Ancestral Barcodes” to better reflect the function of the fCpG barcodes. The manuscript is edited to correct errors, and a new Supplement is added to further explain fCpG barcode mechanics and present new supporting data.

**Reviewer #1 (Public review):**
I thank Reviewer 1 for his constructive comments. Major noted weaknesses were (1) insufficient clarity and brevity of the methodology, (2) inconsistent or erroneous use of neurodevelopmental concepts, and (3) lack of consideration for alternative explanations.

(1) The methodology is now outlined in detailed in a new Supplement, including simulations that indicate that the error rate consistent with the experimental data is about 0.01 changes in methylation per fCpG site per division.

(2) Conceptual and terminology errors noted by the Reviewers are corrected in the manuscript.

(3) I agree completely with the alternative explanation of Reviewer 1 that fCpGs are “a cellular division clock that diverges over 'time'”. Differences between more traditional cell type classifiers and fCpG barcodes are more fully outlined in the new Supplement. Ancestry recorded by fCpGs and cell type classifiers are confounded because cells of the same phenotype typically have common progenitors---cells within a clade have similar fCpG barcodes because they are closely related. fCpG barcodes can compliment cell type classifiers with additional information such as mitotic ages, ancestry within a clade, and daughter cell migration.

**Reviewer #1 (Recommendations for the authors):**
(1) A lot of the interpretations suffer from an extremely loose/erroneous use of developmental concepts and a lack of transparency. For instance:a) The thalamus is not part of the brain stem

Corrected.

b) The pons contains cells other than inhibitory neurons in the data; the same is true for the hippocampus which contains multiple cell types

Corrected to refer to the specific cell types in these regions.

c) The author talks about the rostral-caudal timing a lot which is not really discussed to this degree in the cited references. Thus, it is also unclear how interneurons fit in this model as they are distinguished by a ventral-dorsal difference from excitatory neurons. Also, it is unclear whether the timing is really as distinct as claimed. For instance, inhibitory neurons and excitatory neurons significantly overlap in their birth timing. Finally, conceptually, it does not make sense to go by developmental timing as the author proposes that it is the number of divisions that is relevant. While they are somewhat correlated there are potentially stark differences.

The manuscript attempts to describe what might be broadly expected when barcodes are sampled from different cell types and locations. As a proposed mitotic clock, the fCpG barcode methylation level could time when each neuron ceased division and differentiated. The wide ranges of fCpG barcode methylation of each cell type (Fig 2A) would be consistent with significant overlap between cell types. The manuscript is edited to emphasize overlapping rather than distinct sequential differentiation of the cell types.

d) Neocortical astrocytes and some oligodendrocytes share a lineage, whereas a subset of oligodendrocytes in the cortex shares an origin with interneurons. This could confound results but is never discussed.

The manuscript does not assess glial lineages in detail because neurons were preferentially included in the sampling whereas glial cells were non-systematically excluded. This sampling information is now included in the section “fCpG barcode identification”.

e) Neocortical interneurons should be more closely related in terms of lineage-to-excitatory neurons than other inhibitory neurons of, for instance, the pons. This is not clearly discussed and delineated.

This is not discussed. It may not be possible analyze these details with the current data. The ancestral tree reconstructions indicate that excitatory neurons that appear earlier in development (and are more methylated) are more often more closely related to inhibitory neurons.

f) While there is some spread of excitatory neurons tangentially, there is no tangential migration at the scale of interneurons as (somewhat) suggested/implied here.

The abstract and results have been modified to indicate greater inhibitory than excitatory neuron tangential migration, but that the extent of excitatory neuron tangential migration cannot be determined because of the sparse sampling and that barcodes may be similar by chance.

g) The nature of the NN cells is quite important as cells not derived from the neocortical anlage are unlikely to share a developmental origin (e.g., microglia, endothelial cells). This should be clarified and clearly stated.

The manuscript is modified to indicate that NN cells are microglial and endothelial cells. These cells have different developmental origins, and their data are present in Fig 2A, but are not further used for ancestral analysis.

(2) The presentation is often somewhat confusing to me and lacks detail. For instance:a) The methods are extremely short and I was unable to find a reference for a full pipeline, so other researchers can replicate the work and learn how to use the pipeline.

The pipeline including python code is outlined in the new Supplement

b) Often numbers are given as ~XX when the actual number with some indication of confidence or spread would be more appropriate.

Data ranges are often indicated with the violin plots.

c) Many figure legends are exceedingly short and do not provide an appropriate level of detail.

Figure legends have been modified to include more detail

d) Not defining groups in the figure legends or a table is quite unacceptable to me. I do not think that referring to a prior publication (that does not consistently use these groups anyway) is sufficient.

The cell groups are based on the annotations provided with each single cell in the public databases.

e) The used data should be better defined and introduced (number of cells, different subtypes across areas, which cells were excluded; I assume the latter as pons and hippocampus are only mentioned for one type of neuronal cells, see also above).

The data used are present in Supplemental File 2 under the tab “cell summary H01, H02, H04”.

f) Why were different upper bounds used for filtering for H01 and H02, and H04 is not mentioned? Why are inhibitory and excitatory neurons specifically mentioned (Lines 61-66)?

The filtering is used to eliminate, as much as possible, cell type specific methylation, or CpG sites with skewed neuron methylation. The filtering eliminates CpG sites with high or low methylation within each of the three brains, and within the two major neuron subtypes. The goal is to enrich for CpG sites with polymorphic but not cell type specific methylation. This process is ad hoc as success criteria are currently uncertain. The extent of filtering is balanced by the need to retain sufficient numbers of fCpGs to allow comparisons between the neurons.

g) What 'progenitor' does the author refer to? The Zygote? If yes, can the methylation status be tested directly from a zygote? There is no single progenitor for these cells other than the zygote. Does the assumption hold true when taking this into account? See, for instance, PMID 33737485 for some estimation of lineage bottlenecks.

A brain progenitor cell can be defined as the common ancestor of all adult neurons, and is the first cell where each of its immediate daughter cell lineages yield adult neurons. The zygote is a progenitor cell to all adult cells, and barcode methylation at the start of conception, from the oocyte to the ICM, was analyzed in the new Supplement. The proposed brain progenitor cell with a fully methylated barcode was not yet evident even in the ICM.

(3) I am generally not convinced that the fCpGs represent anything but a molecular clock of cell divisions and that many of the similarities are a function of lower division numbers where the state might be more homogenous. This mainly derives from the issues cited above, the lack of convincing evidence to the contrary, and the sparsity of the assessed data.

Agree that the fCpG barcode is a mitotic clock that becomes polymorphic with divisions. As outlined in the new Supplement, ancestry and cell type are confounded because cells of the same type typically have a common progenitor.

a) There appears little consideration or modeling of what the ability to switch back does to the lineage reconstruction.

fCpG methylation flipping is further analyzed and discussed in the new Supplement.

b) None of the data convinced me that the observations cannot be explained by the aforementioned molecular clock and systematic methylation similarities of cell types due to their cell state.

See above

(4) Uncategorized minor issues:a) The author should explain concepts like 'molecular clock hypothesis' (line 27) or 'radial unit hypothesis' (line 154), as they are somewhat complex and might not be intuitive to readers.

The molecular clock hypothesis is deleted and the radial unit hypothesis is explained in more detail in the manuscript.

b) Line 32: '[...] replication errors are much higher compared to base replication [...]'. I think this is central to the method and should be better explained and referenced. Maybe even through a schematic, as this is a central concept for the entire manuscript.

The fCpG barcode mechanics are better explained in the new Supplement. With simulations, the fCpG flip rate is about 0.01 per division per fCpG.

c) Line 41: 'neonatal'. Does the author mean to say prenatal? Most of the cells discussed are postmitotic before birth.

Corrected to prenatal.

d) Line 96: what does 'flip' mean in this context? Please also see the comment on Figure 2C.

Edited to “chage”

e) Lines 134-135: I am not sure whether the author claims to provide evidence for this question, and I would be careful with claims that this work does resolve the question here.

Have toned down claims as evidence for my analysis is currently inadequate.

f) Lines 192-193: I disagree as the fCpGs can switch back and the current data does not convince me that this is an improvement upon mosaic mutation analysis. In my mind, the main advantage is the re-analysis of existing data and the parallel functional insights that can be obtained.

Lineage analysis is more straightforward with DNA sequencing, but with an error rate of ~10-9 per base per division, one needs to sequence a billion base pairs to distinguish between immediate daughter cells. By contrast, with an inferred error rate of ~10-2 per fCpG per division, much less sequencing (about a million-fold less) is needed to find differences between daughter cells.

g) Lines 208-209: I would be careful with claims of complexity resolution given many of the limitations and inherent systematic similarities, as well as the potential of fCpGs to change back to an ancestral state later in the lineage.

Have modified the manuscript to indicate the analysis would be more challenging due to back changes.

h) There seem to be few figures that assess phenomena across the three brains. Even when they exist there is no attempt to provide any statistical analyses to support the conclusions or permutations to assess outlier status relative to expectations.

The analysis could be more extensive, but with only three brains, any results, like this study itself, would be rightly judged inadequate.

Figure 2B: there appears to be a higher number of '0s' for, for instance, inhibitory neurons compared to excitatory neurons. Is that correct and worth mentioning? The changing axes scales also make it hard to assess.

Inhibitory neurons do appear to have more unmethylated fCpGs compared to excitatory neurons, but in general, most inhibitory fCpGs are methylated with a skew to fully methylated fCpGs, consistent with the barcode starting predominately methylated and inhibitory neurons generally appearing earlier in development relative to excitatory neurons.

j) Figure 2C: I have several issues with this. A minor one is the use of 'Glial' which, I believe, does not appear anywhere else before this, so I am unclear what this curve represents. Generally, however, I am not sure what the y-axis represents, as it is not described in the methods or figure legend. I initially thought it was the cumulative frequency, but I do not think that this squares with the data shown in B. I appreciate the overall idea of having 'earlier'/samples with fewer divisions being shifted to the left, but it is very confusing to me when I try to understand the details of the plot.

This graph is now better described in the legend. “Glial” cells are defined as oligodendrocytes and astrocytes. Other non-neuronal cells (such a microglial cells) have now been removed from the graph.

This graph attempts to illustrate how it may be possible to reconstruct brain development from adult neurons, assuming barcodes are mitotic clocks that become polymorphic with cell division. The X axis is “time”, and the Y axis indicates when different cell types reach their adult levels. The cartoon indicates what is visually present along the X axis during development--- brainstem, then ganglionic eminences with a thin cortex, and finally the mature brain with a robust cortex. Time for the X axis is barcode methylation and starts at 100% and ends at 50% or greater methylation. The fCpG barcode methylation of each cell places it on this timeline and indicates when it ceased dividing and differentiated.

The Y axis indicates the progressive accumulation of the final adult contents of each cell type during this timeline. Early in development, the brain is rudimentary and adult cells are absent. At 90% methylation, only the inhibitory neurons in the pons are present. At 80% methylation, some excitatory neurons are beginning to appear. Inhibitory neurons in the pons have reached their final adult levels and many other inhibitory neuron types are reaching adult levels. By 70% methylation, most inhibitory neurons have reached their adult levels, and more adult excitatory neurons (mainly low cortical neurons, L4-6) and glial cells are beginning to appear. By 60% methylation, inhibitory neurogenesis has largely finished. Adult excitatory neurons and glial cells are more abundant and reach their adult levels by 50% or greater cell barcode methylation levels.

The graph illustrates a rough alignment between mitotic ages inferred by barcode methylation levels and the physical appearances of different neuronal types during development. Many neurons die during development, and this graph, if valid, indicates when neurons that survive to adulthood appear during development.

k) Figure 4Bff: it is confusing to me that the text jumps to these panels after introducing Figure 5. This makes it very hard to read this section of the text.

The Figures appear in the order they are first referred to in the text.

l) Figure 5A: could any of this difference be explained by the shared lineage of excitatory neurons and dorsal neocortical glia?

Not sure

m) Figure 5B: after stating that interneurons have a higher lineage fidelity, the figure legend here states the opposite and I am somewhat confused by this statement.

The legend and text have been clarified. Fig 5A restricts fidelity to within inhibitory cell types. Fig 5B compares between neuron subtypes, and illustrates more apparent inhibitory subtype switching, albeit there are more interneuron subtypes than excitatory subtypes.

n) Figure 5E: generally, the use of tSNE for large pairwise distance analysis is often frowned upon (e.g., PMID 37590228), and I would reconsider this argument.

This analysis was an attempt to illustrate that cells of the same phenotype based on their tSNE metrics can be either closely or more distantly related. Although the tSNE comparisons were restricted to subtypes (and not to the entire tSNE graph), tSNE are not designed for such comparisons. This graph and discussion are deleted.

**Reviewer #2 (Public review):**
The manuscript by Shibata proposed a potentially interesting idea that variation in methylcytosine across cells can inform cellular lineage in a way similar to single nucleotide variants (SNVs). The work builds on the hypothesis that the "replication" of methylcytosine, presumably by DNMT1, is inaccurate and produces stochastic methylation variants that are inherited in a cellular lineage. Although this notion can be correct to some extent, it does not account for other mechanisms that modulate methylcytosines, such as active gain of methylation mediated by DNMT3A/B activity and activity demethylation mediated by TET activity. In some cases, it is known that the modulation of methylation is targeted by sequence-specific transcription factors. In other words, inaccurate DNMT1 activity is only one of the many potential ways that can lead to methylation variants, which fundamentally weakens the hypothesis that methylation variants can serve as a reliable lineage marker. With that being said (being skeptical of the fundamental hypothesis), I want to be as open-minded as possible and try to propose some specific analyses that might better convince me that the author is correct. However, I suspect that the concept of methylation-based lineage tracing cannot be validated without some kind of lineage tracing experiment, which has been successfully demonstrated for scRNA-seq profiling but not yet for methylation profiling (one example is Delgado et al., nature. 2022).

I thank Reviewer 2 for the careful evaluation. The validation experiment example (Delgado et al.) introduced sequence barcodes in mice, which is not generally feasible for human studies.

(1) The manuscript reported that fCpG sites are predominantly intergenic. The author should also score the overlap between fCpG sites and putative regulatory elements and report p-values. If fCpG sites commonly overlap with regulatory elements, that would increase the possibility that these sites being actively regulated by enhancer mechanisms other than maintenance methyltransferase activity.

As mentioned for Reviewer 1, fCpGs are filtered to eliminate cell type specific methylation.

(2) The overlap between fCpG and regulatory sequence is a major alternative explanation for many of the observations regarding the effectiveness of using fCpG sites to classify cell types correctly. One would expect the methylation level of thousands of enhancers to be quite effective in distinguishing cell types based on the published single-cell brain methylome works.

As mentioned above, the manuscript did not clearly indicate that the fCpG barcode is not a cell type classifier. The distinctions between fCpG barcodes and cell type classifiers are better explained in the new Supplement.

(3) The methylation level of fCpG sites is higher in hindbrain structures and lower in forebrain regions. This observation was interpreted as the hindbrain being the "root" of the methylation barcodes and, through "progressive demethylation" produced the methylation states in the forebrain. This interpretation does not match what is known about methylation dynamics in mammalian brains, in particular, there is no data supporting the process of "progressive demethylation". In fact, it is known that with the activation of DNMT3A during early postnatal development in mice or humans (Lister et al., 2013. Science), there is a global gain of methylation in both CH and CG contexts. This is part of the broader issue I see in this manuscript, which is that the model might be correct if "inaccurate mC replication" is the only force that drives methylation dynamics. But in reality, active enzymatic processes such as the activation of DNMT3A have a global impact on the methylome, and it is unclear if any signature for "inaccurate mC replication" survives the de novo methylation wave caused by DNMT3A activity.

Reviewer 2 highlights a critical potential flaw in that any ancestral signal recorded by random replication errors could be overwritten by other active methylation processes. I cannot present data that indicates fCpG replication errors are never overwritten, but new data indicate barcode reproducibility and stability with aging.

New data are also present where barcodes are compared between daughter cells (zygote to ICM) in the setting of active and passive demethylation, when germline methylation is erased. This new analysis shows that daughter cells in 2 to 8 cell embryos have more related barcodes than morula or ICM cells. The subsequent active remethylation by a wave of DNMT3A activity may underlie the observation that the barcode appears to start predominately methylated in brain progenitors.

(3) Perhaps one way the author could address comment 3 is to analyze methylome data across several developmental stages in the same brain region, to first establish that the signal of "inaccurate mC replication" is robust and does not get erased during early postnatal development when DNMT3A deposits a large amount of de novo methylation.

See above

(4) The hypothesis that methylation barcodes are homogeneous among progenitor cells and more polymorphic in derived cells is an interesting one. However, in this study, the observation was likely an artifact caused by the more granular cell types in the brain stem, intermediate granularity in inhibitory cells, and highly continuous cell types in cortical excitatory cells. So, in other words, single-cell studies typically classify hindbrain cell types that are more homogenous, and cortical excitatory cells that are much more heterogeneous. The difference in cell type granularity across brain structures is documented in several whole-brain atlas papers such as Yao et al. 2023 Nature part of the BICCN paper package.

As noted above, fCpG barcode polymorphisms and cell type differentiation are confounded because cells of the same phenotype tend to have common progenitors. The fCpG barcode is not a cell type classifier but more a cell division clock that becomes polymorphic with time. Although fCpG barcodes could be more polymorphic in cortical excitatory cells because there are many more types, fCpG barcodes would inherently become more polymorphic in excitatory cells because they appear later in development.

(5) As discussed in comment 2, the author needs to assess whether the successful classification of cell types (brain lineage) using fCpG was, in fact, driven by fCpG sites overlapping with cell-type specific regulatory elements.

Although unclear in the manuscript, the fCpG is not a cell classifier and the barcode is polymorphic between cells of the same type. fCpG barcodes can appear to be cell classifiers because cell types appear at different times during development, and therefore different cell types have characteristic average barcode methylation levels.

(6) In Figure 5E, the author tried to address the question of whether methylation barcodes inform lineage or post-mitotic methylation remodeling. The Y-axis corresponds to distances in tSNE. However, tSNE involves non-linear scaling, and the distances cannot be interpreted as biological distances. PCA distances or other types of distances computed from high-dimensional data would be more appropriate.

The Figure and discussion are deleted (similar comment by Reviewer 1)

**Reviewer #3 (Public review):**
Summary:In the manuscript entitled "Human Brain Barcodes", the author sought to use single-cell CpG methylation information to trace cell lineages in the human brain.Strengths:Tracing cell lineages in the human brain is important but technically challenging. Lineage tracing with single-cell CpG methylation would be interesting if convincing evidence exists.Weaknesses:As the author noted, "DNA methylation patterns are usually copied between cell division, but the replication errors are much higher compared to base replication". This unstable nature of CpG methylation would introduce significant problems in inferring the true cell lineage. The unreliable CpG methylation status also raises the question of what the "Barcodes" refer to in the title and across this study. Barcodes should be stable in principle and not dynamic across cell generations, as defined in Reference#1. It is not convincing that the "dynamic" CpG methylation fits the "barcodes" terminology. This problem is even more concerning in the last section of results, where CpG would fluctuate in post-mitotic cells.

I thank Reviewer 3 for his thoughtful and careful evaluation. I think the “barcode” terminology is appropriate. Dynamic engineered barcodes such as CRISPR/Cas9 mutable barcodes are used in biology to record changes over time. The fCpG barcode appears to start with a single state in a progenitor cell and changes with cell division to become polymorphic in adult cells. Therefore, I think the description of a dynamic fCpG barcode is appropriate.

**Reviewer #3 (Recommendations for the authors):**
(1) As the author noted, "DNA methylation patterns are usually copied between cell division, but the replication errors are much higher compared to base replication". This unstable nature of CpG methylation would introduce significant problems in inferring the true cell lineage. To establish DNA methylation as a means for lineage tracing, one control experiment would be testing whether the DNA methylation patterns can faithfully track cell lineages for in vitro differentiated & visibly tracked cell lineages. Has this kind of experiment been done in the field?

These types of experiments have not been performed to my knowledge and an appropriate tissue culture model is uncertain. New single cell WGBS data from the zygote to ICM indicate that more immediate daughter cells have more related barcodes even in the setting of active DNA demethylation.

(2) The study includes assumptions that should be backed with solid rationale, supporting evidence, or reference. Here are a couple of examples:a) the author discarded stable CpG sites with <0.2 or >0.8 average methylation without a clear rationale in H02, and then used <0.3 and >0.7 for a specific sample H01.

The filtering was ad hoc and was used to remove, as much as possible, CpG sites with cell type specific or patient specific methylation. CpG sites with skewed methylation are more likely cell type specific, whereas X chromosome CpG sites with methylation closer to 0.5 in male cells are more likely to be unstable. The ad hoc filtering attempted to remove cell specific CpGs sites while still retaining enough CpG sites to allow comparisons between cells.

b) The author assumed that the early-formed brain stem would resemble progenitors better and have a higher average methylation level than the forebrain. However, this difference in DNA methylation status could reflect developmental timing or cell type-specific gene expression changes.

This observation that brain stem neurons that appear early in development have highly methylated fCpG barcodes in all 3 brains supports the idea that the fCpG barcode starts predominately methylated. Alternative explanations are possible.

(3) The conclusion that excitatory neurons undergo tangential migration is unclear - how far away did the author mean for the tangential direction? Lateral dispersion is known, but it would be striking that the excitatory neurons travel across different brain regions. The question is, how would the author interpret shared or divergent methylation for the same cell type across different brain regions?

As noted with Reviewer 1, this analysis is modified to indicate that evidence of tangential migration is greater for inhibitory than excitatory neurons, but the extent of excitatory neuron migration is uncertain because of sparse sampling, and because fCpG barcodes can be similar by chance.

(4) The sparsity and resolution of the single-cell DNA methylation data. The methylation status is detected in only a small fraction (~500/31,000 = 1.6%) of fCpGs per cell, with only 48 common sites identified between cell pairs. Given that the human genome contains over 28 million CpG sites, it is important to evaluate whether these fCpGs are truly representative. How many of these sites were considered "barcodes"?

fCpG barcodes are distinct from traditional cell type classifiers, and how fCpGs are identified are better outlined in the new Supplement.

(5) While focusing on the X-chromosome may simplify the identification of polymorphic fCpGs, the confidence in determining its methylation status (0 or 1) is questionable when a CpG site is covered by only one read. Did the author consider the read number of detected fCpGs in each cell when calculating methylation levels? Certain CpG sites on autosomes may also have sufficient coverage and high variability across cells, meeting the selection criteria applied to X-chromosome CpGs.

In most cases, a fCpG site was covered by only a single read

(6) The overall writing in the Title, the Main text, Figure legends, and Methods sections are overly simplified, making it difficult to follow. For instance, how did the author perform PWD analysis? How did they handle missing values when constructing lineage trees?There is not much introduction to lineage tracing in the human brain or the use of DNA methylation to trace cell lineage.

These shortcomings are improved in the manuscript and with the new Supplement. The analysis pipeline including the Python programs are outlined and included as new Supplemental materials. IQ tree can handle the binary fCpG barcode data and skips missing values with its standard settings.

Line 80: it is unclear: "Brain patterns were similar"

Clarified

Line 98: The meaning is unclear here: "Outer excitatory and glial progenitor cells are present" What are these glial progenitor cells and when/how they stop dividing?

The glial cells are the oligodendrocytes and astrocytes. The main take away point is that these glial cells have low barcode methylation, consistent with their appearances later in development.

Line 104: It is unclear if this is a conclusion or assumption -- "A progenitor cell barcode should become increasingly polymorphic with subsequent divisions." The "polymorphic" happens within the progenitors, their progenies, or their progenies at different time points.

The statement is now clarified as an assumption in the manuscript.

Similarly line 134 "Barcodes would record neuronal differentiation and migration." Is this a conclusion from this study or a citation? How is the migration part supported?

The reasoning is better explained in the manuscript. Migration can be documented if immediate daughter cells with similar barcodes are found in different parts of the adult brain, albeit analysis is confounded by sparse sampling and because barcodes may be similar by chance.

Line 148 and 150: "Nearest neighbor ... neuron pairs" in DNA methylation status would conceivably reflect their cell type-specific gene expression, how did the author distinguish this from cell lineage?

As noted above, because cells with similar phenotypes usually arise from common progenitors, cells within a clade are also usually related. However, the barcodes are still polymorphic within a clade and potentially add complementary information on mitotic ages, ancestry within a clade, and possible cell migration.

Figure 3C: "Cells that emerge early in development" Where are they on the figure?

Hindbrain neurons differentiate early in development and their barcodes are more methylated. The figure has been modified to label some of the values with their neuron types. Also, the older figure mistakenly included data from all 3 brains and now the data are only from brain H01.

Figures 4D and 4E, distinguishing cell subtypes is challenging, as the same color palette is used for both excitatory and inhibitory neurons.

Unfortunate limitations due to complexity and color limitations

Figures 4 and 5, what are these abbreviations?

The abbreviations are presented in Figure 1 and maintained in subsequent figures.